# HiST: Spatial Transcriptomics Prediction via Multi-Level Hyperbolic Representation Learning

## Abstract

Spatial Transcriptomics (ST) merges the benefits of pathology images and gene expression, linking molecular profiles with tissue structure to analyze spot-level function comprehensively. Predicting gene expression from histology images is a cost-effective alternative to expensive ST technologies. However, existing methods mainly focus on spot-level image-to-gene matching but fail to leverage the full hierarchical structure of ST data, especially on the gene expression side, leading to incomplete image-gene alignment. Moreover, a challenge arises from the inherent information asymmetry: gene expression profiles contain more molecular details that may lack salient visual correlates in histological images, demanding a sophisticated representation learning approach to bridge this modality gap. We propose **HiST**, a framework for ST prediction that learns multi-level image-gene representations by modeling the data's inherent hierarchy within hyperbolic space, a natural geometric setting for such structures. First, we design a **Multi-Level Representation Extractor** to capture both spot-level and niche-level representations from each modality, providing context-aware information beyond individual spot-level image-gene pairs. Second, a **Hierarchical Hyperbolic Alignment** module is introduced to unify these representations, performing spatial alignment while hierarchically structuring image and gene embeddings. This alignment strategy enriches the image representations with molecular semantics, significantly improving cross-modal prediction. HiST achieves state-of-the-art performance on three public datasets from different tissues, paving the way for more scalable and accurate spatial transcriptomics prediction.

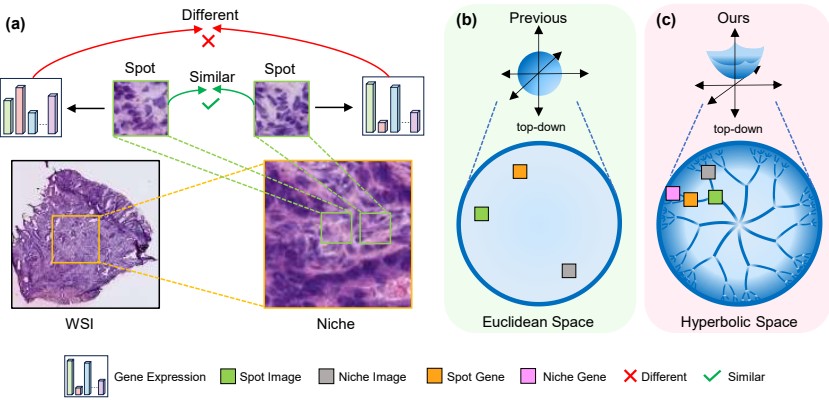

Figure 1: **ST data characteristics**. (a) A WSI contains hierarchical structures and visually similar patterns may correspond to different gene expressions. (b) Other works mainly model ST data in Euclidean Space, which neglects niche-level gene and can lead to biased biological insights. (c) Our hyperbolic approach models hierarchies based on information specificity, where a general concept (image/spot) entails its more specific, information-rich counterpart (gene/niche), enabling more informative representation learning.

## 1 INTRODUCTION

Pathological images, particularly Hematoxylin and Eosin (H&E) stained Whole Slide Images (WSIs), provide critical insights into cell morphology and tissue architecture, serving as a cornerstone in biomedical research and clinical diagnosis (Lu et al., 2024; Chen et al., 2024a; Li et al., 2024). Gene expression data complement these pathological images by elucidating the molecular mechanisms underlying observed features, thereby enhancing disease diagnosis and facilitating therapeutic target identification (Ash et al., 2021). Spatial Transcriptomics (ST) integrates both modalities by capturing spatially resolved gene expression and cellular morphology simultaneously (Ståhl et al., 2016), aligning molecular profiles with tissue structure at micrometer resolution (Williams et al., 2022). Despite its advantages, ST has not achieved widespread clinical adoption due to its high cost and laborious experiment compared to traditional techniques (Zhang et al., 2022; Choe et al., 2023). Consequently, there has been an increasing attention in predicting spatially resolved gene expression directly from pathological images using deep learning approaches (Wang et al., 2025a).

Recent studies have explored diverse strategies for this prediction task, including direct inference from spot-level images (He et al., 2020; Monjo et al., 2022), integration of multi-scale features across WSIs (Chung et al., 2024; Wang et al., 2025b), and contrastive learning to align spot-level images with gene expression profiles (Xie et al., 2023). Although these methods have shown promising results, several critical questions remain underexplored. Our study is motivated by two key questions in ST prediction: (1) *Can integrating broader pathological and genetic context improve spot-level gene expression inference?* Previous studies often primarily utilized multi-scale pathological features for gene expression prediction, neglecting the multi-level structure inherent in gene expression itself (Jaume et al., 2024b; Chen et al., 2024b), which spans cellular and tissue-level scales. In reality, both broader pathological context and bulk genetic programs can significantly influence the gene expression profile at each spot (Chen et al., 2020; Nirmal et al., 2022; Wu et al., 2025; Ye et al., 2024). (2) *How can we effectively learn more target-modality information (i.e., gene expression) during training to enhance cross-modal prediction?* Biological heterogeneity frequently results in visually similar pathology patches exhibiting distinct gene expression patterns (Zhu et al., 2025; Pizurica et al., 2024; Fujii et al., 2024; Tang et al., 2025), as illustrated in Figure 1 (a). This phenomenon indicates that standard image encoders may fail to capture the subtle morphological cues for predicting these molecular variations. Instead of viewing this as an ill-posed one-to-many problem, we contend that the key is to learn a more powerful and molecularly-informed image representation.

To address these two questions, we introduce **HiST**, a novel framework for ST prediction by learning multi-level hyperbolic image-gene representations. HiST tackles these challenges with two core components. First, our **Multi-Level Representation Extractors** capture hierarchical representations from both pathology images and their corresponding gene expression profiles. They extract multimodal information at both spot- level and niche-level, where a niche consists of a central spot and its surrounding neighbors, enabling the capture of comprehensive morphological and molecular patterns across spatial scales. Second, our **Hierarchical Hyperbolic Alignment** module acts as a powerful structural regularizer rather than a generative model. It uses the unique properties of hyperbolic geometry to impose a meaningful inductive bias on the latent space, guiding the model to learn molecularly-informed features.

We define our hierarchical relationships based on **information specificity**. In this view, a concept A entails a concept B if B is a semantically richer and more specific instance of A. For example, the concept of a "dog on a beach" is more specific and information-rich than "dog", and is thus considered the child concept. Following this principle, we establish two key hierarchies in our framework: (1) A spot-level representation entails its context-rich niche-level counterpart. (2) A morphological image entails its corresponding gene expression. This is because the gene profile contains fine-grained molecular information that offers a much more specific description of the tissue's state than the more general pathology image. HiST learns powerful, context-aware representations by modeling information-based hierarchies in hyperbolic space, which is inherently more suited for capturing such structures than Euclidean space (Hsu et al., 2021).

We demonstrate HiST's effectiveness on three public datasets from diverse tissues, where it consistently outperforms state-of-the-art models. Our contributions are summarized as:

- We propose **HiST**, a novel framework for predicting spatially resolved gene expression from WSIs by learning multi-level hyperbolic representations that capture the intrinsic hierarchical structure of ST data.

- We design **Multi-Level Representation Extractors** to capture spot- and niche-level representations from both modalities, providing comprehensive biological insights.

- We introduce **Hierarchical Hyperbolic Alignment** to structurally regularize the latent space, improving cross-modal feature integration.

- Extensive experiments on three public datasets demonstrate that HiST consistently outperforms existing approaches, underscoring its robust efficacy in spatial gene expression prediction.

## 2 RELATED WORK

### 2.1 PREDICTION OF GENE EXPRESSION FROM HISTOLOGY IMAGES

Recent methodologies for predicting spatially resolved gene expression from histology images have advanced through diverse computational paradigms, including ST-Net (He et al., 2020), BLEEP (Xie et al., 2023), TRIPLEX (Chung et al., 2024), and Stem (Zhu et al., 2025). Local image-to-expression regression models like ST-Net employ ResNet50 (He et al., 2016) to directly map H&E image patches to gene expressions. While effective in deterministic prediction, these methods assume injective mappings between morphology and transcription, overlooking biological heterogeneity. Multi-scale integration approaches like TRIPLEX extract and fuse multi-resolution features from WSIs using attention mechanisms. Although these methods capture multi-resolution visual patterns, they lack explicit constraints to preserve the essential biological hierarchy. Generative models like Stem address the uncertainty in expression prediction by generating probabilistic gene expression profiles. While these paradigms better preserve transcriptional variability, they neglect the inherent data hierarchy. In contrast to these prior works, HiST explicitly models the intrinsic parent-child relationships between spots and their surrounding niches across both imaging and gene expression modalities.

### 2.2 MULTIMODAL CONTRASTIVE REPRESENTATION LEARNING

Contrastive learning is a pivotal technique for cross-modal tasks by aligning representations across different modalities. For example, CLIP (Radford et al., 2021) employs contrastive learning to align paired images and texts in a shared Euclidean embedding space. Inspired by CLIP, BLEEP (Xie et al., 2023) adapts contrastive learning to histology and gene expression, using direct interpolation in the embedding space for efficient, decoder-free predictions. These two models rely on Euclidean embeddings, which limit their ability to capture hierarchical relationships. To overcome these limitations, MERU (Desai et al., 2023) embeds image and text into hyperbolic space, leveraging its geometric properties to build a hierarchical representation space through contrastive and entailment losses. Building on MERU, HyCoCLIP (Pal et al., 2024) introduces intra-modal hierarchical modeling by extracting object boxes from images and their corresponding textual descriptions, establishing hierarchical links between box regions and the full image-text pair. HyCoCLIP's dependence on pre-trained object detection models to derive these boxes from given captions may result in potential inaccuracies. In contrast, HiST directly leverages the inherent structure of ST data, from spot-level to niche-level contexts, avoiding uncertainties associated with external feature extraction.

## 3 METHOD

The overview of HiST is illustrated in Figure 2. First, we briefly describe the preliminaries of the hyperbolic geometry in Section 3.1. Second, we present the Multi-Level Representation Extractors in Section 3.2. Third, we introduce the Hierarchical Hyperbolic Alignment in Section 3.3. Finally, we describe the Gene Decoder and our Overall Objective Function in Section 3.4.

### 3.1 PRELIMINARIES

**Hyperbolic Geometry** Hyperbolic Geometry is a fundamental class of non-Euclidean geometry with a constant negative curvature. This distinguishing characteristic results in an exponential growth

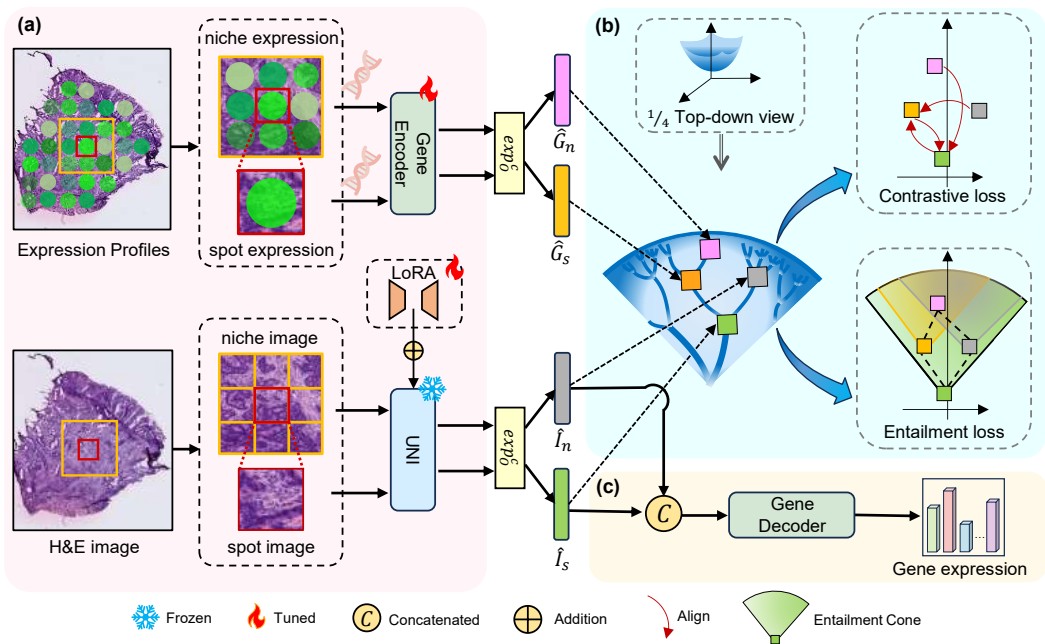

Figure 2: **Overview of HiST.** HiST consists of three components. (a) **Multi-Level Representation Extractors** capture spot- and niche-level features from both images and gene expression. (b) **Hierarchical Hyperbolic Alignment** module projects these features into a shared hyperbolic latent space. It uses contrastive alignment for corresponding image-gene pairs and entailment alignment to structurally regularize the latent space according to information hierarchies. (c) **Gene Decoder** uses the resulting aligned and context-aware image representations to predict spot-level gene expression.

of volume with respect to radius, in stark contrast to Euclidean geometry, which exhibits zero curvature and polynomial volume scaling (Mettes et al., 2024). Consequently, hyperbolic spaces are naturally adept at representing tree-like or hierarchical data structures, where the number of elements increases exponentially with depth (Hsu et al., 2021; Pal et al., 2024). Due to their negative curvature, hyperbolic spaces cannot be isometrically embedded in Euclidean spaces of equivalent dimensionality without compromising distances or angles. To address this issue, several geometric models are employed for their representation and computation, including the Poincaré ball model and the Lorentz model (Cannon et al., 1997; Cho et al., 2022).

**Lorentz Model**   Lorentz model is widely preferred due to its numerical stability and straightforward geodesic calculations (Nickel & Kiela, 2018). The Lorentz model $\mathbb{L}_c^n$ embeds the $n$-dimensional hyperbolic space as the upper sheet of a two-sheeted hyperboloid in $(n + 1)$-dimensional Minkowski space, with a constant curvature $-c < 0$, it consists of all vectors satisfying:

$$\mathbb{L}_c^n = \{\mathbf{x} \in \mathbb{R}^{n+1} : \langle \mathbf{x}, \mathbf{x} \rangle_{\mathbb{L}} = -\frac{1}{c}, x_{time} = \sqrt{1/c + \|\mathbf{x}_{space}\|^2}, c > 0\}, \tag{1}$$

where points $\mathbf{x} \in \mathbb{R}^{n+1}$ in $\mathbb{L}_c^n$ can be represented as $[x_{time}, \mathbf{x}_{space}]$. $x_{time} \in \mathbb{R}$ and $\mathbf{x}_{space} \in \mathbb{R}^n$ denote the *time component* and the *spatial component* (Desai et al., 2023), respectively. For two vectors $\mathbf{x}, \mathbf{y} \in \mathbb{L}_c^n$, the Lorentzian inner product $\langle \cdot, \cdot \rangle_{\mathbb{L}}$ is defined as $\langle \mathbf{x}, \mathbf{y} \rangle_{\mathbb{L}} = \langle \mathbf{x}_{space}, \mathbf{y}_{space} \rangle_{\mathbb{E}} - x_{time} y_{time}$, where $\langle \mathbf{x}, \mathbf{y} \rangle_{\mathbb{E}}$ represents the Euclidean inner product in $\mathbb{R}^n$. Besides, the Lorentzian distance $d_{\mathbb{L}}(\mathbf{x}, \mathbf{y})$ measures the length of the shortest path between two points $\mathbf{x}$ and $\mathbf{y}$, which is formulated as:

$$d_{\mathbb{L}}(\mathbf{x}, \mathbf{y}) = \sqrt{1/c} \cdot \cosh^{-1}(-c\langle \mathbf{x}, \mathbf{y} \rangle_{\mathbb{L}}). \tag{2}$$

**Tangent Space and Exponential Map**   The tangent space of $\mathbf{x} \in \mathbb{L}^n$ is denoted by $\mathcal{T}_{\mathbf{x}}\mathbb{L}_c^n$ which is precisely defined as the set of vectors orthogonal to $\mathbf{x}$ under the Lorentzian inner product:

$$\mathcal{T}_{\mathbf{x}}\mathbb{L}_c^n = \{\mathbf{v} \in \mathbb{R}^{n+1} : \langle \mathbf{x}, \mathbf{v} \rangle_{\mathbf{L}} = 0\}. \tag{3}$$

A fundamental mechanism for connecting the tangent space to the hyperbolic manifold is the exponential map. The exponential map $\exp_{\mathbf{x}}^c : \mathcal{T}_{\mathbf{x}}\mathbb{L}_c^n \to \mathbb{L}_c^n$ projects tangent vector $\mathbf{v}$ onto the $\mathbb{L}_c^n$ along a geodesic emanating from $\mathbf{x}$ in the direction of $\mathbf{v}$, given by:

$$\exp_{\mathbf{x}}^c(\mathbf{v}) = \cosh(\sqrt{c}\|\mathbf{v}\|_{\mathbb{L}})\mathbf{x} + \frac{\sinh(\sqrt{c}\|\mathbf{v}\|_{\mathbb{L}})}{\sqrt{c}\|\mathbf{v}\|_{\mathbb{L}}}\mathbf{v}, \tag{4}$$

where $\|\mathbf{v}\|_{\mathbb{L}} = \sqrt{\langle \mathbf{v}, \mathbf{v}\rangle_{\mathbb{L}}}$ is the Lorentzian norm. Moreover, the exponential map serves as a bridge between Euclidean and hyperbolic geometries. By interpreting Euclidean vectors as tangent vectors at the origin $\mathbf{O} = [\sqrt{1/c}, 0, \ldots, 0] \in \mathbb{R}^{n+1}$ of the hyperbolic space (Mettes et al., 2024; Pal et al., 2024; Khrulkov et al., 2020), we begin by extending the Euclidean embedding $\mathbf{v}_{euc} \in \mathbb{R}^n$ into $\mathbb{R}^{n+1}$ by defining a vector $\mathbf{v} = [0, \mathbf{v}_{euc}] \in \mathbb{R}^{n+1}$. This vector $\mathbf{v}$ is situated in the tangent space at the origin $\mathbf{O}$ of the hyperboloid as $\langle \mathbf{O}, \mathbf{v}\rangle_{\mathbb{L}} = 0$. Thus, $\mathbf{v}$ can be projected onto the hyperboloid $\mathbb{L}_c^n$ employing the exponential map:

$$\mathbf{x}_{space} = \exp_{\mathbf{O}}^c(\mathbf{v}_{euc}) = \frac{\sinh(\sqrt{c}\|\mathbf{v}_{euc}\|_{\mathbb{E}})}{\sqrt{c}\|\mathbf{v}_{euc}\|_{\mathbb{E}}}\mathbf{v}_{euc}. \tag{5}$$

Then we can directly calculate the corresponding time component $x_{time}$ from $\mathbf{x}_{space}$. The detailed derivations of the above equations can be found in Appendix A.

**Hyperbolic Entailment Loss**   The entailment cone $\mathcal{R}_{\mathbf{y}}$ constitutes a region of point $\mathbf{y}$ where all points $\mathbf{x} \in \mathcal{R}_{\mathbf{y}}$ represent child concepts of the parent concept $\mathbf{y}$ (Ganea et al., 2018; Desai et al., 2023), defined by the half-aperture:

$$\text{aper}(\mathbf{y}) = \sin^{-1}\left(\frac{2K}{\sqrt{c}\|\mathbf{y}_{space}\|}\right), \tag{6}$$

where $K = 0.1$ determines boundary conditions near the origin. To enforce the partial order relationship where $\mathbf{y}$ entails $\mathbf{x}$, the penalty is formulated as:

$$\mathcal{L}_{entail}(\mathbf{y}, \mathbf{x}) = \max\left(0, \text{ext}(\mathbf{y}, \mathbf{x}) - \text{aper}(\mathbf{y})\right), \tag{7}$$

where $\text{ext}(\mathbf{y}, \mathbf{x})$ denotes the exterior angle defined as $\text{ext}(\mathbf{y}, \mathbf{x}) = \cos^{-1}\left(\frac{x_{time} + y_{time}c\langle\mathbf{y}, \mathbf{x}\rangle_{\mathbb{L}}}{\|\mathbf{y}_{space}\|\sqrt{(c\langle\mathbf{y}, \mathbf{x}\rangle_{\mathbb{L}})^2 - 1}}\right)$.

### 3.2 Multi-Level Representation Extractors

**Multi-Level Pathological Images Extractor**   Following the previous works (Xie et al., 2023; Zhu et al., 2025), a spot-level image patch $X_s \in \mathbb{R}^{3 \times L_s \times L_s}$ of each identified spot is extracted and preprocessed from a H&E stained image, with the spot positioned at the center of the patch, as depicted in Figure 2 (a), where $L_s$ represents the size of the spot image. While $X_s$ directly corresponds to the target spot gene expression, the additional nearby visual information from high-level pathology patches can significantly contribute to the analysis (Chung et al., 2024; Lin et al., 2024). Therefore, we introduce the niche-level image patch $X_n \in \mathbb{R}^{3 \times L_n \times L_n}$, which is defined as a higher-level region composed of the central spot-level patch $X_s$ and its spatially adjacent spot-level patches. $L_n$ signifies the patch size of the niche image. These neighboring patches are selected based on spatial proximity using the K-Nearest Neighbors (KNN) algorithm. By cropping the region of these patches, $X_n$ forms a larger image region that provides a broader field of view and enhanced contextual information about the surrounding tissue microenvironment.

We leverage UNI (Chen et al., 2024a), a pathology foundation model pre-trained on large-scale histology images, to extract feature embeddings for spot-level and niche-level image patches. As the original UNI was not well-suited for large-sized niche-level image patches, we resized the images in our dataset and fine-tuned UNI using the Low-Rank Adaptation (LoRA) technique (Hu et al., 2022), leading to improved multi-level visual representations. Consider a frozen pre-trained weight matrix $W_{origin} \in \mathbb{R}^{d \times d}$, where $d$ denotes the dimension. The updated weight matrix is formulated as $W_{new} = W_{origin} + \Delta W = W_{origin} + BA$, where the update $\Delta W \in \mathbb{R}^{d \times d}$ expressed by the product of two smaller trainable matrices: $B \in \mathbb{R}^{d \times r}$, $A \in \mathbb{R}^{r \times d}$, and the rank $r \ll d$. This approach enables us to adapt the model to the characteristics of our data while substantially reducing the computational resources required for fine-tuning. The multi-level image representations $I_s \in \mathbb{R}^d$ and $I_n \in \mathbb{R}^d$ are extracted by $I_s, I_n = MIE(X_s, X_n; \theta_{UNI}, \Delta\theta_{lora})$. Here, $MIE$ represent the UNI model adapted by LoRA, $\theta_{uni}$ and $\Delta\theta_{lora}$ denote the frozen parameters of UNI and the trainable parameters of LoRA modules.

**Multi-Level Genomic Profiles Extractor** Let $Y_s \in \mathbb{R}^N$ be the associated spot-level gene expression profile of the spot-level image $X_s$, where $N$ is the gene set size. In the same vein as the niche-level image patch, we introduce the niche-level gene expression profile $Y_n \in \mathbb{R}^N = \frac{1}{|S|} \sum_{i \in S} Y_s^i$, where $S = \{Y_s^1, \cdots, Y_s^K\}$ denotes the expression profile set of $Y_s$ and its neighbors, $K-1$ is the number of selected neighbors. $G_s, G_n = MGE(Y_s, Y_n; \theta_{gene})$, where $MGE$ denotes the multi-level genomic profiles extractor with trainable parameters implemented by a trainable fully connected network $\theta_{gene}$, $G_s \in \mathbb{R}^d$ and $G_n \in \mathbb{R}^d$ are the spot-level and niche-level gene embeddings, respectively.

### 3.3 HIERARCHICAL HYPERBOLIC ALIGNMENT

To obtain better representations for facilitating the subsequent tasks, the alignment is a pivotal method which bridges the gap of different modalities (Xie et al., 2023; Zhang et al., 2025; Li et al., 2021). However, common implementations of alignment, such as BLEEP (Xie et al., 2023) , directly close the different items in Euclidean space, which may not be appropriate for hierarchical data like ST. To address this problem, we design a **Hierarchical Contrastive Alignment** module, which aligns the different modalities at different levels in hyperbolic space. Subsequently, we introduce a **Hierarchical Entailment Alignment** module to regularize the partial order in ST data.

**Hierarchical Contrastive Alignment (HCA)** Using Equation 5, let $\exp_O^c(\cdot) : \mathbb{R}^d \to \mathbb{L}_c^d$ map Euclidean features to hyperbolic space with trainable curvature $-c < 0$ and origin $O$. This yields hyperbolic spatial components $\{\hat{I}_s^{space}, \hat{I}_n^{space}, \hat{G}_s^{space}, \hat{G}_n^{space}\} = \exp_O^c (\{I_s, I_n, G_s, G_n\})$, while the corresponding time components can be calculated by Equation 1. The hyperbolic representations $\hat{I}_s$, $\hat{I}_n$, $\hat{G}_s$ and $\hat{G}_n$ are obtained by concatenating spatial components and time components. To align the spot-level image embedding to the spot-level embedding, we employ modified infoNCE loss (Oord et al., 2018), in which the cosine similarity is replaced by the Lorentzian distance $d_{\mathbb{L}}(\cdot, \cdot)$ described in Equation 2. The contrastive loss is defined as follows:

$$\mathcal{L}_{align}(\hat{I}_s, \hat{G}_s) = -\frac{1}{B} \sum_{i=1}^{B} \log \frac{\exp(d_{\mathbb{L}}(\hat{I}_s^i, \hat{G}_s^i)/\tau)}{\sum_{j=1, j \neq i}^{B} \exp(d_{\mathbb{L}}(\hat{I}_s^i, \hat{G}_s^j)/\tau)}, \quad (8)$$

where $B$ denotes the batch size and $\tau$ is the temperature parameter. To better utilize in-batch negatives, we also align spot-level gene to spot-level image embeddings using $\mathcal{L}_{align}(\hat{G}_s, \hat{I}_s)$. Since spot-level features represent more general characteristics, a single spot-level feature may correspond to multiple niche-level features within a batch. To avoid such undesirable negative alignment, we only consider the alignment from niche-level features to spot-level features, i.e., $\mathcal{L}_{align}(\hat{G}_n, \hat{I}_s)$ and $\mathcal{L}_{align}(\hat{I}_n, \hat{G}_s)$. The objective function of Hierarchical Contrastive Alignment can be expressed as:

$$\mathcal{L}_{HCA} = \frac{1}{4}(\mathcal{L}_{align}(\hat{I}_s, \hat{G}_s) + \mathcal{L}_{align}(\hat{G}_s, \hat{I}_s) + \mathcal{L}_{align}(\hat{G}_n, \hat{I}_s) + \mathcal{L}_{align}(\hat{I}_n, \hat{G}_s)) \quad (9)$$

**Hierarchical Entailment Alignment (HEA)** Beyond spot-niche hierarchies, we account for the non-identical nature of image features and gene features. We recognize that gene features provide finer-grained molecular insights. Thus, we posit that gene features are the child concept of images in hyperbolic space. In our ST data, this hierarchy can be summarized as spot-level features entailing niche-level features, and pathological images entailing their corresponding gene expression profiles. In order to directly constrain this hierarchical structure, we leverage Hyperbolic Entailment Loss $\mathcal{L}_{entail}(\cdot, \cdot)$ described in Equation 7. Therefore, the final objective function of this module is formulated as:

$$\mathcal{L}_{HEA} = \frac{1}{4}(\mathcal{L}_{entail}(\hat{I}_s, \hat{I}_n) + \mathcal{L}_{entail}(\hat{G}_s, \hat{G}_n) + \mathcal{L}_{entail}(\hat{I}_s, \hat{G}_s) + \mathcal{L}_{entail}(\hat{I}_n, \hat{G}_n)). \quad (10)$$

### 3.4 GENE DECODER BASED ON ALIGNED REPRESENTATIONS AND OBJECTIVE FUNCTION

To predict the spot-level gene expression profiles, we directly concatenate the aligned representations ( $I_s$ and $I_n$) and feed the result into a gene decoder implemented by Multi-Layer Perceptron (MLP) (LeCun et al., 2015), which can be expressed by $Y^{pred} = Decoder_{gene}(\text{concat}(I_s, I_n))$. MSE loss is leveraged to optimize this decoder: $\mathcal{L}_{pred} = \|Y^{pred} - Y_s\|_2^2$.

Table 1: Performance comparison on three spatial transcriptomics datasets. Higher values on PCC@10, PCC@50, PCC@200 are better. Lower values on MAE, MSE are better.

| Dataset | Model | PCC@10 ↑ | PCC@50 ↑ | PCC@200 ↑ | MSE ↓ | MAE ↓ |
|---|---|---|---|---|---|---|
| Colorectum | TRIPLEX | 0.701±0.128 | 0.624±0.154 | 0.462±0.191 | 1.869±0.803 | 1.056±0.239 |
| | StNet | 0.646±0.134 | 0.570±0.142 | 0.419±0.176 | 1.686±0.373 | 1.023±0.134 |
| | BLEEP | 0.637±0.112 | 0.556±0.120 | 0.382±0.160 | 2.038±0.587 | 1.096±0.164 |
| | Stem | 0.670±0.116 | 0.573±0.130 | 0.399±0.166 | 1.788±0.418 | 1.032±0.138 |
| | HiST(Ours) | **0.721±0.105** | **0.642±0.128** | **0.477±0.184** | **1.498±0.456** | **0.958±0.158** |
| Skin | TRIPLEX | 0.831±0.094 | 0.799±0.114 | 0.740±0.142 | 0.981±0.466 | 0.685±0.205 |
| | StNet | 0.804±0.105 | 0.779±0.117 | 0.726±0.140 | 0.993±0.469 | 0.689±0.198 |
| | BLEEP | 0.788±0.111 | 0.761±0.123 | 0.704±0.145 | 1.117±0.540 | 0.701±0.221 |
| | Stem | 0.782±0.094 | 0.748±0.113 | 0.687±0.138 | 1.276±0.703 | 0.730±0.261 |
| | HiST(Ours) | **0.839±0.086** | **0.812±0.102** | **0.758±0.129** | **0.932±0.418** | **0.657±0.182** |
| Kidney | TRIPLEX | 0.579±0.095 | 0.485±0.084 | 0.351±0.066 | 1.122±0.204 | 0.855±0.104 |
| | StNet | 0.523±0.105 | 0.435±0.095 | 0.305±0.064 | 1.167±0.217 | 0.847±0.078 |
| | BLEEP | 0.518±0.112 | 0.434±0.102 | 0.310±0.071 | 1.233±0.244 | 0.865±0.085 |
| | Stem | 0.535±0.111 | 0.414±0.084 | 0.271±0.059 | 1.380±0.347 | 0.911±0.115 |
| | HiST(Ours) | **0.617±0.094** | **0.526±0.088** | **0.390±0.070** | **1.077±0.155** | **0.817±0.058** |

**The Training Objective Function** The training objective of HiST is twofold: (1) to align pathological image and gene expression profiles across multiple levels by modeling the hierarchical structure of ST data, and (2) to accurately predict gene expression from image features alone. Ultimately, this objective function consists of two components: hierarchical alignment loss and ST prediction loss, defined by:

$$\mathcal{L} = \mathcal{L}_{pred} + \alpha(\mathcal{L}_{HCA} + \beta\mathcal{L}_{HEA}), \tag{11}$$

where $\alpha$ balances the loss components, and $\beta$ controls the entailment loss effect.

# 4 EXPERIMENTS AND RESULTS

## 4.1 EXPERIMENTAL SETTINGS

**Dataset** To evaluate HiST, we collected three public datasets from the HEST-1K dataset (Jaume et al., 2024a), a high-quality collection of spatial transcriptomics data with standardized processing and rich metadata. (1) Colorectum dataset (Valdeolivas et al., 2024) comprises 14 WSIs (0.45 $\mu$m per pixel) with a total of 20,733 spots; (2) Skin (Schäbitz et al., 2022) includes 46 WSIs and over 35,008 spots. The resolution of pathology images is about 0.52 $\mu$m/pixel; (3) Kidney (Lake et al., 2023) provides 23 WSIs and 25,944 spots at a resolution of approximately 0.76 $\mu$m/pixel.

**ST Preprocessing** To account for variations in image resolution across datasets, we adopted a physics-aware patch extraction strategy rather than using fixed pixel dimensions for image cropping. Specifically, we calculated the patch size for each spot based on its physical diameter and crop the corresponding images at their respective resolutions to obtain spot-level image patches. The niche-level patch is created by cropping the region encompassing the central spot and its K-nearest neighbors (determined by spatial coordinates). Subsequently, all extracted patches are resized to a uniform 224×224 pixel resolution. For the gene expression data, we select the top 200 Highly Mean, Highly Variant Genes (HMHVG). Gene expression counts for each spot were subsequently log-transformed.

**Evaluation Protocol** To ensure robust model evaluation, we performed five independent random splits of the WSI samples for each dataset, allocating 80% for training, 10% for validation, and 10% for testing in each iteration. The exact WSI IDs used for each of the five splits are provided in our code to ensure full reproducibility. Our evaluation metrics include top-$k$ mean Pearson Correlation Coefficient (PCC@$k$), mean squared error (MSE), and mean absolute error (MAE), similar to (Zhu et al., 2025; Chung et al., 2024).

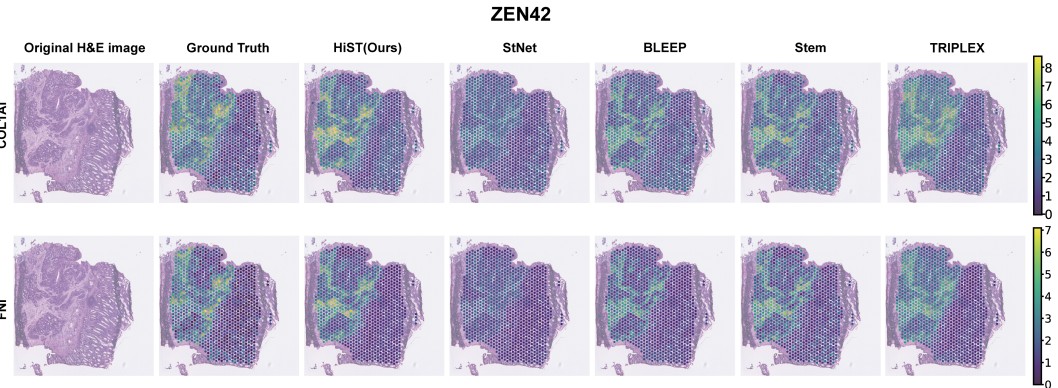

Figure 3: Visualization of the spatial distribution of the COL1A1 (Top) gene and FN1 gene (Bottom) in ZEN42 sample.

## 4.2 EXPERIMENTAL RESULTS AND VISUALIZATION

**Baseline Comparison**    Table 1 shows that HiST outperforms all existing methods across all datasets (Colorectum, Skin, and Kidney), highlighting HiST's superior accuracy and robustness in predicting gene expression from pathology images across diverse biological contexts, which were trained under a rigorous, fair comparison protocol (details in Appendix B.2). For instance, compared to the second-best method TRIPLEX, HiST achieves an improvement of 6.33% and 3.24% in PCC@200 on the Kidney and Skin datasets, respectively. Notably, TRIPLEX wachieves the next-best performance, underscoring the crucial role of multi-level structure in accurately predicting gene expression. Acknowledging the variance in the results of Table 1 , we performed paired t-tests in Appendix C.1 to confirm the statistical significance of HiST's improvements. To further demonstrate the robustness of our model, we also conducted a series of additional experiments, including an efficiency benchmark (Appendix C.2), cross-laboratory generalization tests (Appendix C.3), robustness analysis on the HVG gene set (Appendix C.4), evaluation with patient-level data splitting (Appendix C.5) and so on.

**Biomarker Visualization**    To further qualitatively assess the model's behavior, we performed visualization of sample ZEN42 in the Colorectum dataset, focusing on two established colorectal cancer biomarkers, i.e., COL1A1 (Zhang et al., 2018; Pawlak et al., 2025) and FN1 (Sun et al., 2020). Figure 3 demonstrates that HiST more accurately captures the key high-expression regions compared to other methods. More visualizations are available in Section D of Appendix.

**Clinical Downstream Task Validation**    We designed a downstream validation experiment to further validate the clinical utility of the representations learned by HiST. First, we employed our model, pre-trained on the Colorectum dataset, to perform zero-shot inference on H&E slides from an independent external dataset, TCGA-COADREAD (colon and rectal adenocarcinoma). The inferred gene expression profiles were then used to train a Random Forest classifier for predicting microsatellite instability (MSI) status, a critical clinical biomarker for immunotherapy response (Feng et al., 2024). As shown in Table 2, we note that applying some baseline models to this large-scale experiment was computationally infeasible. **TRIPLEX**'s global attention mechanism led to prohibitive memory require-

Table 2: Performance on MSI Status Classification (AUROC).

| Model | AUROC | |
|---|---|---|
| | MSI-H | MSS |
| TRIPLEX | - | - |
| StNet | 0.57±0.08 | 0.54±0.03 |
| BLEEP | 0.55±0.05 | 0.53±0.06 |
| Stem | - | - |
| Ours | **0.72±0.06** | **0.60±0.06** |

ments (>60GB of VRAM for a single WSI), while **Stem**'s diffusion-based inference was excessively slow (over 230 hours for the entire cohort). The gene profiles predicted by HiST led to significantly better MSI prediction performance compared to other baselines. This result indicates that the representations learned by HiST generalize remarkably well and carry tangible clinical value, underscoring its potential for downstream clinical applications.

## 4.3 ABLATION STUDY

We performed an ablation study on the model's structure and hyperparameters to observe the strategy of alignment, input data for gene decoder and the impact of LoRA. Here, we describe the results in Colorectum dataset. The ablation study on the impact of LoRA, as well as more experiment results on other datasets, can be found in Appendix E.

**Strategy of Alignment**   We compared the different alignment strategies including: a) removing only the gene-image regularization term of the HEA loss (w/o G-I HEA), b) removing the entire HEA loss (w/o HEA), c) removing the entire Hierarchical Hyperbolic Alignment (HHA) module (w/o HEA + HCA), d) replacing the HHA module by a MERU variant in Hyperbolic Space (without mutil-level representation learning) (Desai et al., 2023) and e) replacing the HHA module by a CLIP variant in Euclidean Space (Radford et al., 2021), as shown in Table 3. The results imply that our approach can learn better representations by leveraging inherent hierarchies, enhancing the overall model performance. Notably, the performance gap between our full model and its Euclidean counterpart (CLIP) strongly validates our core hypothesis on the superiority of hyperbolic space for this task.

Table 3: Ablation study of the alignment strategy.

| Alignment | PCC@10 ↑ | PCC@50 ↑ | PCC@200 ↑ | MSE ↓ | MAE ↓ |
|---|---|---|---|---|---|
| w/o G-I HEA | 0.708±0.116 | 0.633±0.139 | 0.470±0.186 | 1.523±0.471 | 0.972±0.160 |
| w/o HEA | 0.708±0.118 | 0.627±0.142 | 0.465±0.186 | 1.523±0.444 | 0.969±0.154 |
| w/o HEA + HCA | 0.697±0.141 | 0.615±0.162 | 0.456±0.201 | 1.675±0.810 | 0.997±0.250 |
| MERU | 0.705±0.101 | 0.618±0.159 | 0.451±0.142 | 1.535±0.341 | 0.970±0.125 |
| CLIP | 0.693±0.093 | 0.605±0.102 | 0.441±0.135 | 1.523±0.219 | 1.011±0.124 |
| **Ours** | **0.721±0.105** | **0.642±0.128** | **0.477±0.184** | **1.498±0.456** | **0.958±0.158** |

**Input of Decoder**   We evaluated the impact of different input strategies on the decoder's performance, including using a) only spot-level image, b) only niche-level images and c) the combination of both. As shown in Figure 4, the results reveal that our combined approach yields the best performance across all metrics. In summary, this ablation study confirms that the integration of both spot-level and niche-level information is critical for achieving optimal performance.

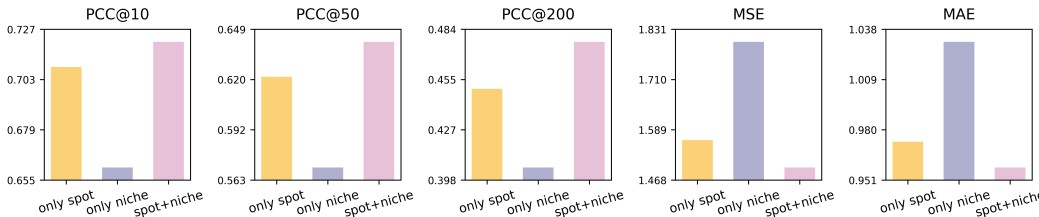

Figure 4: Ablation study of the input data of decoder.

## 5 CONCLUSION

We present HiST, a novel framework that leverages multi-level hyperbolic representations to predict spatial transcriptomics from histology images. By modeling the intrinsic hierarchical structure of ST data within hyperbolic space, HiST learns more comprehensive spatial histological and genetic features. Our comprehensive experimental evaluation demonstrates that HiST consistently outperforms state-of-the-art approaches, underscoring the potential of geometric deep learning in spatial omics analysis. We hope this framework inspires future research toward geometric-aware multimodal learning in the biological domain, harnessing the inherent geometry of biological systems for designing more sophisticated representation models.

## ETHICS STATEMENT

All data utilized in this study, including Whole Slide Images and their corresponding gene expression profiles, were sourced from publicly available and properly cited datasets. No new data involving human subjects was collected for this research, and all data was fully anonymized, containing no personally identifiable information. Our work fully complies with the licensing and terms of use for all original data sources.

## REPRODUCIBILITY STATEMENT

We are committed to ensuring the reproducibility of this research. The code, data details, and implementation specifics required to reproduce our experimental results are as follows:

**Code Availability** The complete source code and experiment scripts are available at the following anonymous repository: `https://anonymous.4open.science/r/12116-V2`.

**Dataset and Preprocessing** Our research is based on publicly available datasets. Their detailed descriptions, sources, and the full data preprocessing pipeline (including patch extraction and gene selection) are elaborated in Section 4.1 and Appendix B.1.

**Implementation Details** Specifics of the model implementation, including the hyperparameter settings for all experiments, and the detailed training procedure, are provided in Appendix B.

We believe this information is sufficient to support the verification and extension of our work.

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

# Appendix

## CONTENTS

## A  EXPONENTIAL MAP DERIVATIONS

In this section, we present the derivation of the exponential map (Equation 5) in our approach Desai et al. (2023). Let the vector $\mathbf{v} = [0, \mathbf{v}_{euc}] \in \mathbb{R}^{n+1}$ denotes the extension of the Euclidean embedding $\mathbf{v}_{euc} \in \mathbb{R}^n$. This vector belongs in the tangent space at the origin $\mathbf{O} = [\sqrt{1/c}, 0, \ldots, 0] \in \mathbb{R}^{n+1}$ of the hyperboloid as the Lorentzian inner product of these two vectors is zero, where $-c < 0$ is the curvature of the hyperboloid. Based on Equation 4, we can simplify the exponential map by considering only the space components:

$$\mathbf{x}_{space} = \cosh(\sqrt{c}\|\mathbf{v}\|_{\mathbb{L}})0 + \frac{\sinh(\sqrt{c}\|\mathbf{v}\|_{\mathbb{L}})}{\sqrt{c}\|\mathbf{v}\|_{\mathbb{L}}}\mathbf{v}_{euc}, \tag{12}$$

where the first term of this equation is zero. The Lorentzian norm of $\mathbf{v}$ is equal to the Euclidean norm of space components:

$$\|\mathbf{v}\|_{\mathbb{L}} = \sqrt{\langle \mathbf{v}, \mathbf{v} \rangle_{\mathbb{L}}} = \sqrt{0 + \langle \mathbf{v}, \mathbf{v} \rangle_{\mathbb{E}}} = \|\mathbf{v}_{euc}\|, \tag{13}$$

where $\langle \cdot, \cdot \rangle_{\mathbb{E}}$ denotes standard Euclidean inner product. Therefore, this exponential map can be formulated as:

$$\mathbf{x}_{space} = \exp_{\mathbf{O}}^c(\mathbf{v}_{euc}) = \frac{\sinh(\sqrt{c}\|\mathbf{v}_{euc}\|_{\mathbb{E}})}{\sqrt{c}\|\mathbf{v}_{euc}\|_{\mathbb{E}}}\mathbf{v}_{euc}. \tag{14}$$

## B  IMPLEMENTATION DETAILS

### B.1  IMPLEMENTATION DETAILS FOR GENE SELECTION

The gene expression profiles of Spatial Transcriptomics (ST) typically contain approximately 20,000 to 30,000 genes, most of which exhibit low variability. Directly feeding all these genes into a deep learning model would lead to the severe curse of dimensionality. To address this issue, we implemented two gene selection strategies based on previous work (Zhu et al., 2025): 1) The top 200 genes with both high mean expression and high variability (MHHVG); 2) The top 300 genes selected from all highly variable genes (HVG) ranked by mean expression level. Specifically, for each WSI, we extracted the top 2,000 highly variable genes based on its corresponding gene expression profile. These highly variable gene sets are then pooled across all WSIs to form a union set. From this union set, we rank genes by mean expression and variance independently to identify the top 300 genes in each category. The top 300 genes with the highest mean expression are designated as highly expressed genes (HEG), and the top 300 genes with the highest variance are designated as highly variable genes (HVG). We then take the intersection of these two sets to define the Highly Mean and Highly Variable Genes (HMHVG). This approach ensures a robust and consistent gene selection for downstream analyses. Figure 5 displays the Highly Mean and Highly Variable Genes (HMHVG) set, while Figure 6 shows the selected Highly Variable Genes (HVG) set.

### B.2  IMPLEMENTATION DETAILS FOR EXPERIMENTS

We compare HiST against four state-of-the-art methods: TRIPLEX (Chung et al., 2024), StNet (He et al., 2020), BLEEP (Xie et al., 2023), and Stem (Zhu et al., 2025). To ensure a fair and rigorous comparison, we re-implemented all baseline models and trained them from scratch using the exact same data splits described in our evaluation protocol of Section 4.1. For each method, we meticulously followed the hyperparameter tuning strategies outlined in their original publications, adapting their official public code where available. This standardized setup guarantees that all performance differences can be attributed to model architecture and learning strategy rather than variations in data or implementation.

Our model is trained with AdamW with an initial learning rate of 0.0001. The batch size is 128 and the hidden embedding channel is 1024. The hyperparameters of $\alpha$ and $\beta$ are 0.2 and 0.4 respectively. As for LoRA, we only adapt the attention weights of the foundational model with rank 4, applying it to the last 11 attention layers. Besides, we implement mixed precision training using PyTorch's AMP for accelerated computation, with all experiments seeded at 42 for reproducibility. The epochs of training are up to 200 with early stopping, and we assume models have converged when the validation loss fails to improve for 10 consecutive epochs. All experiments are trained on RTX4090 GPUs.

| Dataset | Genes to be predicted |
|---|---|
| Colorectum | 'A2M', 'ACTA2', 'AEBP1', 'AGR2', 'AHNAK', 'ANXA11', 'APOE', 'ASS1', 'ATP1B1', 'ATP5ME', 'ATP6V0C', 'B2M', 'BCAP31', 'BGN', 'BST2', 'BTF3', 'C15orf48', 'C19orf33', 'C1QA', 'C1QB', 'C1R', 'C1S', 'C3', 'CALD1', 'CALM1', 'CCN2', 'CD24', 'CD44', 'CD55', 'CD59', 'CD74', 'CD81', 'CD99', 'CDH17', 'CEACAM5', 'CEACAM6', 'CEACAM7', 'CKB', 'CLCA1', 'CLDN3', 'CLDN4', 'CLDN7', 'COL12A1', 'COL18A1', 'COL1A1', 'COL1A2', 'COL3A1', 'COL4A1', 'COL4A2', 'COL5A1', 'COL5A2', 'COL6A1', 'COL6A2', 'COL6A3', 'COMMD6', 'COX5A', 'COX5B', 'COX6B1', 'COX7B', 'CRIP1', 'CSTB', 'CTSB', 'CTSC', 'CTSD', 'CXCL1', 'CXCL10', 'CXCL14', 'DBI', 'DCN', 'DEFA5', 'DMBT1', 'DUOX2', 'DUSP1', 'EEF1A1', 'EGR1', 'EIF5A', 'ELF3', 'ENO1', 'EPCAM', 'FABP1', 'FAM3D', 'FBLN1', 'FCGBP', 'FLNA', 'FN1', 'FOS', 'FSTL1', 'FTH1', 'FTL', 'FXYD3', 'GPRC5A', 'GPX2', 'GREM1', 'GSN', 'HLA-A', 'HLA-B', 'HLA-C', 'HLA-DPA1', 'HLA-DPB1', 'HLA-DQB1', 'HLA-DRA', 'HLA-DRB1', 'HMGN2', 'HNRNPH1', 'HNRNPU', 'HSPB1', 'HSPD1', 'ID1', 'IDO1', 'IER2', 'IER3', 'IFI30', 'IFI6', 'IGFBP4', 'IGFBP5', 'IGFBP7', 'IGHA1', 'IGHG1', 'IGHG3', 'IGHG4', 'IGHM', 'IGKC', 'IGLC1', 'IGLC2', 'IGLC3', 'IL32', 'IRF1', 'ISG15', 'ITLN1', 'ITM2B', 'ITM2C', 'JCHAIN', 'JUNB', 'JUND', 'JUP', 'KDELR2', 'KLF5', 'KRT18', 'KRT19', 'KRT8', 'LCN2', 'LDHB', 'LGALS1', 'LGALS3', 'LGALS4', 'LUM', 'LY6E', 'LYZ', 'MALAT1', 'MARCKSL1', 'MDK', 'MGP', 'MMP1', 'MMP11', 'MMP2', 'MUC1', 'MUC12', 'MUC13', 'MUC2', 'MUC5B', 'MYL12B', 'MYL9', 'NBL1', 'NDUFA13', 'NORAD', 'NR4A1', 'OLFM4', 'PABPC1', 'PERP', 'PGK1', 'PHGR1', 'PI3', 'PIGR', 'PLA2G2A', 'POSTN', 'PPIB', 'PRSS8', 'PSMB8', 'PTMA', 'RBM3', 'REG1A', 'REG3A', 'REG4', 'RHOA', 'RHOC', 'ROMO1', 'RRBP1', 'S100A14', 'S100A6', 'S100A8', 'S100A9', 'S100P', 'SAT1', 'SELENOP', 'SERPINA1', 'SERPING1', 'SFN', 'SLC12A2', 'SMIM22', 'SNRPB', 'SOD2', 'SPARC', 'SPINK1', 'SPINK4', 'SPINT2', 'SPTBN1', 'ST14', 'STAT1', 'SULF1', 'TAGLN', 'TFF1', 'TFF3', 'TGFBI', 'THBS1', 'THY1', 'TIMP1', 'TIMP2', 'TM9SF2', 'TMEM176B', 'TMEM54', 'TMEM59', 'TMSB4X', 'TPM2', 'TPT1', 'TSPAN1', 'TSPAN3', 'TSPAN8', 'TSPO', 'TYMP', 'UBD', 'UQCR11', 'UQCRH', 'VIM', 'YWHAB', 'ZFP36', 'ZG16' |
| Skin | 'ACTB', 'ACTG1', 'AHNAK', 'ANXA1', 'ANXA2', 'APRT', 'AQP3', 'ARPC2', 'ASPRV1', 'ATP1B3', 'ATP5F1A', 'ATP5F1B', 'ATP5F1E', 'ATP5MC2', 'ATP5MC3', 'ATP5ME', 'ATP5MF', 'ATP5MG', 'B2M', 'BTF3', 'C19orf33', 'CALM1', 'CALML3', 'CALML5', 'CASP14', 'CCL27', 'CD24', 'CD44', 'CD74', 'CD9', 'CDSN', 'CFL1', 'CHCHD2', 'CLTB', 'CNBP', 'CNFN', 'COL17A1', 'COL1A1', 'COL1A2', 'COL3A1', 'COL6A1', 'COL6A2', 'COX4I1', 'COX5B', 'COX6A1', 'COX6B1', 'COX6C', 'COX7A2', 'COX7B', 'COX7C', 'COX8A', 'CRABP2', 'CSNK1A1', 'CST3', 'CST6', 'CSTA', 'CSTB', 'CTNNBIP1', 'CTSB', 'CTSD', 'CXCL14', 'DBI', 'DCD', 'DCN', 'DEFB4A', 'DEGS1', 'DMKN', 'DSC1', 'DSC3', 'DSG1', 'DSG3', 'DSP', 'DSTN', 'DYNLL1', 'EEF1A1', 'EEF1B2', 'EEF1D', 'EEF2', 'EIF1', 'EIF3E', 'EIF3F', 'EIF4G2', 'EIF5A', 'ELOB', 'ENO1', 'FABP5', 'FADS2', 'FAM25A', 'FAU', 'FLG', 'FLG2', 'FTH1', 'FTL', 'GAPDH', 'GJA1', 'GJB2', 'GLTP', 'GPNMB', 'GSTP1', 'GUK1', 'H3-3A', 'H3-3B', 'HINT1', 'HLA-B', 'HLA-C', 'HLA-DPA1', 'HLA-DPB1', 'HLA-DQB1', 'HLA-DRA', 'HLA-DRB1', 'HLA-E', 'HMGB1', 'HNRNPA1', 'HNRNPA2B1', 'HNRNPK', 'HOPX', 'HSP90AA1', 'HSP90AB1', 'HSPA8', 'HSPB1', 'IFI27', 'IFI6', 'IFITM3', 'IGKC', 'ITM2B', 'IVL', 'KLF5', 'KLK7', 'KRT1', 'KRT10', 'KRT14', 'KRT16', 'KRT17', 'KRT2', 'KRT5', 'KRT6A', 'KRT6B', 'KRT6C', 'KRTDAP', 'LAD1', 'LCE1B', 'LCE3D', 'LDHA', 'LGALS1', 'LGALS3', 'LGALS7', 'LGALS7B', 'LMNA', 'LY6D', 'LYPD3', 'LYZ', 'MIF', 'MUCL1', 'MYL6', 'NACA', 'NCCRP1', 'NDUFA4', 'NDUFS5', 'NOP53', 'NPM1', 'OAZ1', 'P4HB', 'PABPC1', 'PERP', 'PFDN5', 'PFN1', 'PGAM1', 'PI3', 'PKM', 'PKP1', 'PLP2', 'POLR2L', 'PPDPF', 'PPIA', 'PPL', 'PRDX1', 'PSAP', 'PSMA7', 'PTMA', 'RAB11A', 'RAC1', 'RACK1', 'RAN', 'RBM3', 'ROMO1', 'RTN4', 'S100A10', 'S100A11', 'S100A14', 'S100A16', 'S100A2', 'S100A4', 'S100A6', 'S100A7', 'S100A8', 'S100A9', 'SBSN', 'SCGB2A2', 'SDC1', 'SELENOW', 'SERBP1', 'SERF2', 'SERPINB3', 'SERPINB4', 'SERPINB5', 'SFN', 'SH3BGRL3', 'SLC25A3', 'SLC25A5', 'SLC25A6', 'SLC2A1', 'SLPI', 'SLURP1', 'SNRPD2', 'SPARC', 'SPINK5', 'SPINT2', 'SPRR1B', 'SPRR2A', 'SPRR2B', 'SPRR2D', 'SPRR2E', 'SPRR2G', 'SUB1', 'TACSTD2', 'TAGLN2', 'TMA7', 'TMBIM6', 'TMEM45A', 'TMSB10', 'TMSB4X', 'TOMM7', 'TPI1', 'TPT1', 'TRIM29', 'TSPO', 'TUBA1B', 'TUBA1C', 'TUBB4B', 'TXN', 'TYMP', 'UBA52', 'UBB', 'UBC', 'UBL5', 'UQCR10', 'UQCR11', 'UQCRB', 'UQCRQ', 'VIM', 'YBX1', 'YBX3', 'YWHAB', 'YWHAZ' |
| Kidney | 'A2M', 'ACADVL', 'ACTA2', 'ACTB', 'ACTG1', 'ADGRG1', 'ADIRF', 'AEBP1', 'ALDOB', 'ANPEP', 'ANXA2', 'APLP2', 'APOE', 'APP', 'AQP1', 'AQP2', 'ASAH1', 'ASS1', 'ATP1A1', 'ATP1B1', 'ATP5F1A', 'ATP5F1D', 'ATP5MC3', 'ATP5ME', 'ATP5MF', 'ATP6V0C', 'B2M', 'BCAM', 'BGN', 'BSG', 'C1QA', 'C1R', 'C7', 'CA2', 'CALB1', 'CALD1', 'CALM1', 'CALM2', 'CANX', 'CD151', 'CD24', 'CD74', 'CD81', 'CD9', 'CDH16', 'CDKN1C', 'CFL1', 'CHCHD10', 'CHCHD2', 'CIRBP', 'CKB', 'CLCNKB', 'CLU', 'COL1A2', 'COL3A1', 'COL4A1', 'COL4A2', 'COX5A', 'COX5B', 'COX6A1', 'COX6B1', 'COX6C', 'COX7A2', 'COX7B', 'COX7C', 'COX8A', 'CRIM1', 'CRYAB', 'CST3', 'CTSB', 'CTSH', 'CXCL12', 'CXCL14', 'CYSTM1', 'DCN', 'DDX17', 'DDX5', 'DEFB1', 'DSTN', 'DUSP1', 'DYNLL1', 'EEF1A1', 'EEF1D', 'EEF1G', 'EEF2', 'EFHD1', 'EIF3K', 'EIF4A1', 'ENG', 'EPAS1', 'EZR', 'FABP1', 'FAU', 'FLNA', 'FTH1', 'FTL', 'FXYD2', 'FXYD4', 'GABARAP', 'GATM', 'GHITM', 'GPX3', 'GSTM3', 'GSTP1', 'GTF2I', 'HINT1', 'HLA-A', 'HLA-B', 'HLA-C', 'HLA-DPA1', 'HLA-DRA', 'HLA-DRB1', 'HLA-E', 'HNRNPA1', 'HNRNPA2B1', 'HSD11B2', 'HSP90AB1', 'HSPA8', 'HSPB1', 'HTRA1', 'IDH2', 'IFITM2', 'IFITM3', 'IGFBP2', 'IGFBP4', 'IGFBP5', 'IGFBP7', 'IGHA1', 'IGHG1', 'IGHG3', 'IGHG4', 'IGKC', 'IGLC1', 'IGLC2', 'IGLC3', 'ITGA3', 'ITGB1', 'ITM2B', 'IVNS1ABP', 'KCNJ1', 'KCNJ15', 'KNG1', 'LAMP1', 'LAMTOR5', 'LAPTM4A', 'LDHA', 'LGALS1', 'LRP2', 'LUM', 'MAL', 'MALAT1', 'MGP', 'MGST1', 'MGST3', 'MIF', 'MIOX', 'MMP7', 'MUC1', 'MYL12A', 'MYL6', 'MYL9', 'MZT2B', 'NAT8', 'NDRG1', 'NDUFA1', 'NDUFA13', 'NDUFA2', 'NDUFA4', 'NDUFA6', 'NDUFB2', 'NDUFB7', 'NDUFB8', 'NDUFB9', 'NDUFC1', 'NDUFS5', 'NDUFS6', 'NDUFV1', 'NEAT1', 'NME2', 'NPC2', 'OAZ1', 'OGDHL', 'OST4', 'P4HB', 'PCBP1', 'PCK1', 'PDZK1IP1', 'PEBP1', 'PEPD', 'PFN1', 'PGK1', 'PHPT1', 'PIGR', 'PODXL', 'POLR2L', 'PPP1R1A', 'PTGDS', 'PTH1R', 'REN', 'RHCG', 'RHOA', 'RNASE1', 'ROMO1', 'RTN4', 'S100A10', 'S100A2', 'S100A6', 'SAT1', 'SDC1', 'SELENOP', 'SERPINA1', 'SERPINA5', 'SFRP1', 'SLC12A1', 'SLC12A3', 'SLC13A3', 'SLC25A3', 'SLC25A5', 'SLC25A6', 'SLC3A1', 'SLC5A12', 'SMIM24', 'SNHG25', 'SOD1', 'SOD2', 'SPARC', 'SPINK1', 'SPP1', 'SRP14', 'SSR4', 'SUCLG1', 'TAGLN', 'TAGLN2', 'THY1', 'TIMP1', 'TIMP2', 'TIMP3', 'TINAGL1', 'TMA7', 'TMSB10', 'TMSB4X', 'TPI1', 'TPM1', 'TPT1', 'TSPAN1', 'TUBB', 'TXN', 'UBA52', 'UGT2B7', 'UMOD', 'UQCRB', 'UQCRC1', 'UQCRFS1', 'VIM', 'WFDC2' |

Figure 5: HMHVG gene selection in each dataset

| Dataset | Genes to be predicted |
|---|---|
| Colorectum | 'A2M', 'ACTA2', 'AEBP1', 'AGR2', 'AHNAK', 'ANXA11', 'ANXA5', 'APLP2', 'APOE', 'ARPC2', 'ASS1', 'ATP1B1', 'ATP5F1A', 'ATP5IF1', 'ATP5ME', 'ATP6V0B', 'ATP6V0C', 'B2M', 'BCAP31', 'BGN', 'BST2', 'BTF3', 'C15orf48', 'C19orf33', 'C1QA', 'C1QB', 'C1R', 'C1S', 'C3', 'CALD1', 'CALM1', 'CALM3', 'CAPZB', 'CCN2', 'CCND1', 'CD24', 'CD44', 'CD55', 'CD59', 'CD74', 'CD81', 'CD99', 'CDH17', 'CEACAM5', 'CEACAM6', 'CEACAM7', 'CHCHD10', 'CIRBP', 'CKB', 'CLCA1', 'CLDN3', 'CLDN4', 'CLDN7', 'CLTA', 'COL12A1', 'COL18A1', 'COL1A1', 'COL1A2', 'COL3A1', 'COL4A1', 'COL4A2', 'COL5A1', 'COL5A2', 'COL6A1', 'COL6A2', 'COL6A3', 'COMMD6', 'COX5A', 'COX5B', 'COX6B1', 'COX7B', 'CRIP1', 'CSTB', 'CTSB', 'CTSC', 'CTSD', 'CTSS', 'CXCL1', 'CXCL10', 'CXCL14', 'CYCS', 'DBI', 'DCN', 'DEFA5', 'DMBT1', 'DUOX2', 'DUSP1', 'EEF1A1', 'EGR1', 'EIF3K', 'EIF5A', 'ELF3', 'ENO1', 'EPCAM', 'FABP1', 'FAM3D', 'FBLN1', 'FCGBP', 'FLNA', 'FN1', 'FOS', 'FSTL1', 'FTH1', 'FTL', 'FXYD3', 'GAS5', 'GLUL', 'GNB1', 'GPI', 'GPRC5A', 'GPX2', 'GREM1', 'GSN', 'HLA-A', 'HLA-B', 'HLA-C', 'HLA-DPA1', 'HLA-DPB1', 'HLA-DQB1', 'HLA-DRA', 'HLA-DRB1', 'HMGN1', 'HMGN2', 'HNRNPAB', 'HNRNPH1', 'HNRNPU', 'HSPB1', 'HSPD1', 'HSPG2', 'ID1', 'IDO1', 'IER2', 'IER3', 'IFI30', 'IFI6', 'IGFBP4', 'IGFBP5', 'IGFBP7', 'IGHA1', 'IGHG1', 'IGHG3', 'IGHG4', 'IGHM', 'IGKC', 'IGLC1', 'IGLC2', 'IGLC3', 'IL32', 'IRF1', 'ISG15', 'ITLN1', 'ITM2B', 'ITM2C', 'JCHAIN', 'JTB', 'JUNB', 'JUND', 'JUP', 'KDELR2', 'KLF5', 'KRT18', 'KRT19', 'KRT8', 'KRTCAP2', 'LCN2', 'LDHB', 'LGALS1', 'LGALS3', 'LGALS4', 'LUM', 'LY6E', 'LYZ', 'MALAT1', 'MARCKSL1', 'MDK', 'MGP', 'MGST1', 'MLEC', 'MMP1', 'MMP11', 'MMP2', 'MORF4L2', 'MUC1', 'MUC12', 'MUC13', 'MUC2', 'MUC5B', 'MYL12B', 'MYL9', 'NAMPT', 'NBL1', 'NDUFA13', 'NDUFB1', 'NDUFS6', 'NOP53', 'NORAD', 'NR4A1', 'OLFM4', 'PABPC1', 'PDIA4', 'PERP', 'PFKL', 'PGK1', 'PHGR1', 'PI3', 'PIGR', 'PLA2G2A', 'POLR2L', 'POSTN', 'PPIB', 'PRDX2', 'PRDX6', 'PRR13', 'PRSS8', 'PSMA1', 'PSMA4', 'PSMB8', 'PTBP1', 'PTGES3', 'PTMA', 'PTMS', 'QSOX1', 'RAB1A', 'RBM3', 'REG1A', 'REG3A', 'REG4', 'RHOA', 'RHOB', 'RHOC', 'RNASE1', 'RNASEK', 'ROMO1', 'RRBP1', 'S100A14', 'S100A4', 'S100A6', 'S100A8', 'S100A9', 'S100P', 'SAT1', 'SDC4', 'SEC61A1', 'SELENOP', 'SELENOW', 'SERBP1', 'SERPINA1', 'SERPING1', 'SFN', 'SKP1', 'SLC12A2', 'SMIM22', 'SNRPB', 'SOD2', 'SPARC', 'SPINK1', 'SPINK4', 'SPINT2', 'SPTBN1', 'SRP14', 'SRRM2', 'SRSF3', 'SS3', 'ST14', 'STAT1', 'SULF1', 'TAGLN', 'TFF1', 'TFF3', 'TGFBI', 'THBS1', 'THY1', 'TIMP1', 'TIMP2', 'TM9SF2', 'TMED10', 'TMEM176B', 'TMEM54', 'TMEM59', 'TMSB4X', 'TPM2', 'TPT1', 'TSPAN1', 'TSPAN3', 'TSPAN8', 'TSPO', 'TST', 'TXNDC5', 'TYMP', 'UBD', 'UQCR10', 'UQCR11', 'UQCRH', 'VIM', 'XBP1', 'YWHAB', 'ZFAS1', 'ZFP36', 'ZFP36L1', 'ZG16' |
| Skin | 'ACTB', 'ACTG1', 'ACTN4', 'AHNAK', 'AHNAK2', 'ANXA1', 'ANXA2', 'APOE', 'APRT', 'AQP3', 'ARPC2', 'ASPRV1', 'ATP1B3', 'ATP5F1A', 'ATP5F1B', 'ATP5F1E', 'ATP5MC2', 'ATP5MC3', 'ATP5ME', 'ATP5MF', 'ATP5MG', 'ATP5PD', 'B2M', 'BTF3', 'BTG1', 'C19orf33', 'C4orf3', 'CALM1', 'CALM2', 'CALML3', 'CALML5', 'CASP14', 'CCL27', 'CD24', 'CD44', 'CD63', 'CD74', 'CD81', 'CD9', 'CDSN', 'CFL1', 'CHCHD2', 'CLTB', 'CNBP', 'CNFN', 'COL17A1', 'COL1A1', 'COL1A2', 'COL3A1', 'COL6A1', 'COL6A2', 'COX4I1', 'COX5B', 'COX6A1', 'COX6B1', 'COX6C', 'COX7A2', 'COX7B', 'COX7C', 'COX8A', 'CRABP2', 'CSDE1', 'CSNK1A1', 'CST3', 'CST6', 'CSTA', 'CSTB', 'CTNNBIP1', 'CTSB', 'CTSD', 'CXCL14', 'DBI', 'DCD', 'DCN', 'DDX5', 'DEFB4A', 'DEGS1', 'DMKN', 'DSC1', 'DSC3', 'DSG1', 'DSG3', 'DSP', 'DSTN', 'DYNLL1', 'EEF1A1', 'EEF1B2', 'EEF1D', 'EEF2', 'EIF1', 'EIF3E', 'EIF3F', 'EIF3K', 'EIF4G2', 'EIF5A', 'ELOB', 'EMP2', 'ENO1', 'EZR', 'FABP5', 'FADS2', 'FAM25A', 'FAU', 'FLG', 'FLG2', 'FTH1', 'FTL', 'GAPDH', 'GJA1', 'GJB2', 'GLTP', 'GPNMB', 'GPX4', 'GRN', 'GSN', 'GSTP1', 'GUK1', 'H3-3A', 'H3-3B', 'HINT1', 'HLA-B', 'HLA-C', 'HLA-DPA1', 'HLA-DPB1', 'HLA-DQB1', 'HLA-DRA', 'HLA-DRB1', 'HLA-E', 'HMGB1', 'HMGN2', 'HNRNPA1', 'HNRNPA2B1', 'HNRNPK', 'HOPX', 'HSP90AA1', 'HSP90AB1', 'HSP90B1', 'HSPA8', 'HSPB1', 'IFI27', 'IFI6', 'IFITM3', 'IGFBP7', 'IGKC', 'ITM2B', 'IVL', 'KLF4', 'KLF5', 'KLK7', 'KRT1', 'KRT10', 'KRT14', 'KRT16', 'KRT17', 'KRT2', 'KRT5', 'KRT6A', 'KRT6B', 'KRT6C', 'KRTDAP', 'LAD1', 'LAPTM4A', 'LCE1B', 'LCE3D', 'LDHA', 'LGALS1', 'LGALS3', 'LGALS7', 'LGALS7B', 'LMNA', 'LY6D', 'LYPD3', 'LYZ', 'MIF', 'MUCL1', 'MYL12A', 'MYL6', 'MZT2B', 'NACA', 'NAP1L1', 'NCCRP1', 'NDUFA1', 'NDUFA4', 'NDUFB1', 'NDUFB4', 'NDUFC1', 'NDUFS5', 'NOP53', 'NPM1', 'OAZ1', 'OST4', 'P4HB', 'PABPC1', 'PCBP1', 'PCBP2', 'PERP', 'PFDN5', 'PFN1', 'PGAM1', 'PGK1', 'PI3', 'PKM', 'PKP1', 'PLP2', 'POLR2L', 'PPDPF', 'PPIA', 'PPIB', 'PPL', 'PPP1R4B1', 'PRDX1', 'PSAP', 'PSMA7', 'PTMA', 'RAC1', 'RAB11A', 'RACK1', 'RAN', 'RBM3', 'RHOA', 'ROMO1', 'RTN4', 'S100A10', 'S100A11', 'S100A14', 'S100A16', 'S100A2', 'S100A4', 'S100A6', 'S100A7', 'S100A8', 'S100A9', 'SBSN', 'SCGB2A2', 'SDC1', 'SELENOW', 'SEM1', 'SERBP1', 'SERF2', 'SERPINB3', 'SERPINB4', 'SERPINB5', 'SFN', 'SH3BGRL3', 'SKP1', 'SLC25A3', 'SLC25A5', 'SLC25A6', 'SLC2A1', 'SLC38A2', 'SLPI', 'SLURP1', 'SNRPD2', 'SPARC', 'SPINK5', 'SPINT2', 'SPRR1B', 'SPRR2A', 'SPRR2B', 'SPRR2D', 'SPRR2E', 'SPRR2G', 'SRP14', 'SSR4', 'SUB1', 'TACSTD2', 'TAGLN2', 'TMA7', 'TMBIM6', 'TMEM45A', 'TMSB10', 'TMSB4X', 'TOMM7', 'TPI1', 'TPT1', 'TRIM29', 'TSPO', 'TUBA1B', 'TUBA1C', 'TUBB', 'TUBB4B', 'TXN', 'TXNIP', 'TYMP', 'UBA52', 'UBB', 'UBC', 'UBE2D3', 'UBL5', 'UQCR10', 'UQCR11', 'UQCRB', 'UQCRQ', 'VIM', 'YBX1', 'YBX3', 'YWHAB', 'YWHAZ', 'ZFP36L1', 'ZFP36L2' |
| Kidney | 'A2M', 'ACADVL', 'ACAT1', 'ACO2', 'ACTA2', 'ACTB', 'ACTG1', 'ADGRG1', 'ADI1', 'ADIRF', 'AEBP1', 'ALDOB', 'ANAPC16', 'ANPEP', 'ANXA2', 'ANXA5', 'APLP2', 'APOE', 'APP', 'AQP1', 'AQP2', 'ARHGDIA', 'ASAH1', 'ASS1', 'ATP1A1', 'ATP1B1', 'ATP5F1A', 'ATP5F1D', 'ATP5MC3', 'ATP5ME', 'ATP5MF', 'ATP6AP2', 'ATP6V0C', 'ATP6V1F', 'B2M', 'BCAM', 'BGN', 'BSG', 'C1QA', 'C1R', 'C7', 'CA2', 'CALB1', 'CALD1', 'CALM1', 'CALM2', 'CANX', 'CAPN2', 'CD151', 'CD24', 'CD74', 'CD81', 'CD9', 'CDH16', 'CDKN1C', 'CFL1', 'CHCHD10', 'CHCHD2', 'CIRBP', 'CKB', 'CLCNKB', 'CLTC', 'CLU', 'COL1A2', 'COL3A1', 'COL4A1', 'COL4A2', 'COX5A', 'COX5B', 'COX6A1', 'COX6C', 'COX7A2', 'COX7B', 'COX7C', 'COX8A', 'CRIM1', 'CRIP2', 'CRYAB', 'CSDE1', 'CSRP1', 'CST3', 'CTSB', 'CTSH', 'CXCL12', 'CXCL14', 'CYSTM1', 'DCN', 'DDT', 'DDX17', 'DDX5', 'DEFB1', 'DSTN', 'DUSP1', 'DYNLL1', 'DYNLL2', 'EEF1A1', 'EEF1D', 'EEF1G', 'EEF2', 'EFHD1', 'EIF3K', 'EIF4A1', 'EIF4A2', 'EIF4B', 'ENG', 'EPAS1', 'EZR', 'FABP1', 'FAU', 'FCGRT', 'FLNA', 'FTH1', 'FTL', 'FXYD2', 'FXYD4', 'GABARAP', 'GATM', 'GHITM', 'GPX3', 'GSN', 'GSTM3', 'GSTP1', 'GTF2I', 'HINT1', 'HLA-A', 'HLA-B', 'HLA-C', 'HLA-DPA1', 'HLA-DRA', 'HLA-DRB1', 'HLA-E', 'HNRNPA1', 'HNRNPA2B1', 'HNRNPH1', 'HSD11B2', 'HSP90AB1', 'HSPA8', 'HSPB1', 'HSPD1', 'HTRA1', 'IDH2', 'IFITM2', 'IFITM3', 'IGFBP2', 'IGFBP4', 'IGFBP5', 'IGFBP7', 'IGHA1', 'IGHG1', 'IGHG3', 'IGHG4', 'IGKC', 'IGLC1', 'IGLC2', 'IGLC3', 'ITGA3', 'ITGB1', 'ITM2B', 'IVNS1ABP', 'JUND', 'KCNJ1', 'KCNJ15', 'KNG1', 'LAMB2', 'LAMP1', 'LAMTOR5', 'LAPTM4A', 'LDHA', 'LGALS1', 'LITAF', 'LRP2', 'LUM', 'MAL', 'MALAT1', 'METTL7A', 'MGP', 'MGST1', 'MGST3', 'MIF', 'MIOX', 'MMP7', 'MUC1', 'MYL12A', 'MYL6', 'MYL9', 'MZT2B', 'NAT8', 'NDRG1', 'NDUFA1', 'NDUFA13', 'NDUFA2', 'NDUFA4', 'NDUFA6', 'NDUFB2', 'NDUFB7', 'NDUFB8', 'NDUFB9', 'NDUFC1', 'NDUFS5', 'NDUFS6', 'NDUFV1', 'NEAT1', 'NME2', 'NOP53', 'NORAD', 'NPC2', 'NUCB1', 'OAZ1', 'OGDHL', 'OST4', 'P4HB', 'PCBP1', 'PCK1', 'PDZK1IP1', 'PEBP1', 'PEPD', 'PFKL', 'PFN1', 'PGAM1', 'PGK1', 'PHPT1', 'PIGR', 'PODXL', 'POLR2L', 'PPIB', 'PPP1R1A', 'PTGDS', 'PTH1R', 'RABAC1', 'RAC1', 'REN', 'RHCG', 'RHOA', 'RNASE1', 'ROMO1', 'RTN4', 'S100A10', 'S100A2', 'S100A6', 'SAT1', 'SCNN1A', 'SCP2', 'SDC1', 'SELENOM', 'SELENOP', 'SERPINA1', 'SERPINA5', 'SFRP1', 'SH3BGRL3', 'SLC12A1', 'SLC12A3', 'SLC13A3', 'SLC25A3', 'SLC25A5', 'SLC25A6', 'SLC3A1', 'SLC5A12', 'SMIM24', 'SNHG25', 'SOD1', 'SOD2', 'SPARC', 'SPINK1', 'SPP1', 'SRP14', 'SSR4', 'ST13', 'SUCLG1', 'SUMO2', 'TAGLN', 'TAGLN2', 'TAPBP', 'THY1', 'TIMP1', 'TIMP2', 'TIMP3', 'TINAGL1', 'TMA7', 'TMSB10', 'TMSB4X', 'TPI1', 'TPM1', 'TPM3', 'TPT1', 'TRIR', 'TSC22D1', 'TSPAN1', 'TUBA1A', 'TUBB', 'TXN', 'UBA52', 'UGT2B7', 'UMOD', 'UQCRB', 'UQCRC1', 'UQCRFS1', 'VIM', 'WFDC2', 'ZFP36L2' |

Figure 6: HVG gene selection in each dataset

B.3 Implementation Details for Ablation Studies

In this work, HiST consists of three main modules: Hierarchical Hyperbolic Alignment (HHA), Gene Decoder and Multi-Level Representation Extractors. To validate the necessity of each component, we design a series of ablation experiments for each module respectively.

**Strategy of Alignment**   We design five strategies of alignment to evaluate the necessity of the HHA module. First, we remove only the gene-image regularization term of the HEA loss (w/o G-I HEA) to investigate the impact of lacking the entailment loss between gene and image. Second, we remove the Hierarchical Entailment Alignment component of HHA (w/o HEA) to investigate the impact of lacking hierarchical constraints in hyperbolic space on model performance. Third, we eliminate HHA module (w/o HEA + HCA), retaining only multi-scale image information to assess its contribution. Fourth, we replace this module with MERU (Desai et al., 2023) (MERU), preserving only the cross-model hierarchical structure while disregarding intra-modal multi-scale information in ST data. Finally, we substitute this module with CLIP (Radford et al., 2021) (CLIP), aligning only spot-level gene expression with spot-level images. These experiments collectively highlight the critical roles of hierarchical constraints, multi-scale information integration, and cross-modal alignment mechanisms in the module's effectiveness.

**Input of Decoder**   In order to investigate the impact of input data on the overall performance of the Gene Decoder, we conducted ablation experiments where we separately use the representations learned from spot-level images and niche-level images as inputs to the Gene Decoder to predict genes.

**Choice of LoRA**   To investigate the significance of the LoRA (Low-Rank Adaptation) component within the Multi-Level Representation Extractors, we design experiments to evaluate its role in efficient finetuning the pre-trained model to current task for multi-scale inputs, particularly niche-level images, which may differ in resolution from standard inputs. By adjusting the number of last attention layers adapted by LoRA, we can control the extent of fine-tuning. Setting the number of adapted attention layers to 0 (i.e., freezing all pre-trained weights) allowed us to establish a baseline where no fine-tuning occurs.

B.4 Implementation Details for Evaluation Metrology

We evaluate model performance using the top-k mean Pearson Correlation Coefficient (PCC@k), mean squared error (MSE), and mean absolute error (MAE). For the $j$-th gene at the $i$-th spot, the PCC of the $j$-th gene ($PCC_j$) is formulated as:

$$PCC_j = \frac{\sum_{i=1}^{n}(\hat{y}_{i,j} - \bar{\hat{y}}_{\cdot,j})(y_{i,j} - \bar{y}_{\cdot,j})}{\sqrt{\sum_{i=1}^{n}(\hat{y}_{i,j} - \bar{\hat{y}}_{\cdot,j})^2}\sqrt{\sum_{i}^{n}(y_{i,j} - \bar{y}_{\cdot,j})^2}}, \tag{15}$$

where $\hat{y}_{i,j}$ and $y_{i,j}$ represent the predicted and actual gene expression of the $j$-th gene at the $i$-th spot, respectively, and $\bar{\hat{y}}_{\cdot,j}$ and $\bar{y}_{\cdot,j}$ denote the mean predicted and actual gene expression of the $j$-th gene across spots. $m$ and $n$ are the numbers of genes and spots, separately. For PCC@k, the average value across top-k $PCC_j$ is calculated as:

$$PCC@k = \frac{1}{k}\sum_{j \in Topk} PCC_j. \tag{16}$$

Subsequently, MSE and MAE can be defined as:

$$MAE = \frac{1}{n \times m}\sum_{i=1}^{n}\sum_{j=1}^{m}|y_{i,j} - \hat{y}_{i,j}|, \tag{17}$$

$$MSE = \frac{1}{n \times m}\sum_{i=1}^{n}\sum_{j=1}^{m}(y_{i,j} - \hat{y}_{i,j})^2. \tag{18}$$

## C  ADDITIONAL EXPERIMENT RESULTS

### C.1  STATISTICAL SIGNIFICANCE ANALYSIS

To assess the statistical significance of the observed performance improvements, we conducted paired t-tests across 5 independent runs. As shown in Table 4, HiST shows statistically significant improvements ($p < 0.05$) on the vast majority of metrics when compared to baselines. For the comparison against TRIPLEX, although a few metrics did not reach the significance threshold, HiST still demonstrated a consistent performance trend rather than an incidental improvement. We attribute these few instances of non-significance to the high biological variance inherent in the datasets, yet the overall results strongly support the robustness of our method.

Table 4: HiST performance improvement (P-value) compared to baseline models on three datasets for HMHVGs.

| Dataset | Model | PCC@10 ↑ | PCC@50 ↑ | PCC@200 ↑ | MSE ↓ | MAE ↓ |
|---|---|---|---|---|---|---|
| Colorectum | TRIPLEX | 2.83% (0.119) | 2.94% (0.135) | 3.24% (0.083) | 19.84% (0.048) | 9.26% (0.034) |
| | StNet | 11.62% (0.012) | 12.62% (0.010) | 13.8% (0.019) | 11.12% (<0.01) | 6.31% (<0.01) |
| | BLEEP | 13.15% (<0.01) | 15.47% (<0.01) | 24.7% (<0.01) | 26.48% (<0.01) | 12.57% (<0.01) |
| | Stem | 7.62% (0.030) | 12.10% (<0.01) | 19.52% (<0.01) | 16.19% (0.015) | 7.13% (0.031) |
| Skin | TRIPLEX | 0.91% (0.057) | 1.59% (0.018) | 2.52% (<0.01) | 5.08% (0.066) | 5.32% (<0.01) |
| | StNet | 4.34% (<0.01) | 4.24% (<0.01) | 4.52% (<0.01) | 6.22% (0.081) | 4.73% (<0.01) |
| | BLEEP | 6.48% (<0.01) | 6.63% (<0.01) | 7.76% (<0.01) | 16.59% (<0.01) | 6.29% (<0.01) |
| | Stem | 7.26% (<0.01) | 8.43% (<0.01) | 10.35% (<0.01) | 27.00% (<0.01) | 10.07% (<0.01) |
| Kidney | TRIPLEX | 6.63% (<0.001) | 8.30% (<0.01) | 10.95% (<0.01) | 3.94% (0.051) | 4.41% (0.043) |
| | StNet | 17.94% (<0.01) | 20.77% (<0.01) | 27.85% (<0.01) | 7.71% (<0.01) | 3.58% (<0.01) |
| | BLEEP | 19.20% (<0.01) | 21.16% (<0.01) | 25.68% (<0.01) | 12.59% (<0.01) | 5.54% (<0.01) |
| | Stem | 15.43% (<0.01) | 26.85% (<0.01) | 43.98% (<0.01) | 21.90% (<0.01) | 10.31% (<0.001) |

### C.2  EFFICIENCY BENCHMARK

We evaluated the computational overhead of our proposed model by comparing the memory usage and runtime performance of HiST against other Euclidean models. While HiST's hyperbolic computations incur higher overhead without specific hardware optimization, we find this acceptable given its superior performance advantages (as shown in Table 1 in the main text), its generalization power, and its future scalability potential for biological modeling. Table 5 details the training memory, training time per epoch, and inference time per epoch on the HCC dataset for 200 target genes.

Table 5: Memory usage and runtime performance comparison of HiST and baseline models on the HCC dataset with 200 target genes.

| Model | Memory Usage (GB) ↓ | Training Time (s/epoch) ↓ | Inference Time (s/epoch) ↓ |
|---|---|---|---|
| TRIPLEX | 13.34 | 30 | 2 |
| StNet | 18.44 | 27 | 1 |
| BLEEP | 5.89 | 10 | 1 |
| Stem | 12.96 | 22 | 1553 |
| HiST (Ours) | 18.39 | 74 | 3 |

### C.3  CROSS-LABORATORY GENERALIZATION

We conducted a cross-laboratory validation experiment to assess the model's generalization capability across different laboratory settings and potential stain variations. We used a Whole Slide Image (WSI) (NCB1563) from a new, independent kidney dataset (Canela et al., 2023). As shown in Table 6, our model, HiST (Ours), achieved significantly superior performance compared to the baseline models. This result highlights HiST's robustness and its ability to capture transferable biological features that generalize beyond the training domain.

Table 6: Performance comparison on a cross-laboratory whole slide image.

| Model | PCC@10 ↑ | PCC@50 ↑ | PCC@200 ↑ | MSE ↓ | MAE ↓ |
|---|---|---|---|---|---|
| TRIPLEX | 0.374±0.056 | 0.265±0.035 | 0.162±0.030 | 1.595±0.121 | 1.015±0.038 |
| StNet | 0.215±0.039 | 0.111±0.033 | -0.003±0.031 | 1.758±0.183 | 1.045±0.046 |
| BLEEP | 0.281±0.042 | 0.199±0.053 | 0.096±0.056 | 1.646±0.215 | 1.019±0.057 |
| Stem | 0.357±0.015 | 0.254±0.008 | 0.145±0.007 | 1.822±0.143 | 1.033±0.042 |
| **HiST** | **0.510±0.016** | **0.361±0.022** | **0.197±0.024** | **1.574±0.128** | **0.983±0.039** |

## C.4 Robustness on HVG Gene Set

We evaluated HiST on the HVG gene set, as described in Section B.1, to further demonstrate the robustness of our model. As presented in Table 7, HiST outperforms other methods, achieving PCC@200 values of 0.505, 0.821, and 0.450 across all three datasets ( Colorectum (Valdeolivas et al., 2024), Skin (Schäbitz et al., 2022), Kidney (Lake et al., 2023)), respectively. Similar to the results for HMHVG in Table 1, TRIPLEX also achieves the second-best overall performance, demonstrating that the integration of multi-scale image features consistently enhances model performance across different gene selection criteria.

Table 7: Performance comparison on the HVG gene set of three spatial transcriptomics datasets. Higher values on PCC@10, PCC@50, PCC@200 are better. Lower values on MAE and MSE are better.

| Dataset | Model | PCC@10 ↑ | PCC@50 ↑ | PCC@200 ↑ | MSE ↓ | MAE ↓ |
|---|---|---|---|---|---|---|
| Colorectum | TRIPLEX | 0.685±0.154 | 0.613±0.181 | 0.484±0.238 | 1.830±0.826 | 1.042±0.257 |
| | StNet | 0.656±0.122 | 0.578±0.138 | 0.447±0.181 | 1.642±0.427 | 1.009±0.154 |
| | BLEEP | 0.662±0.119 | 0.568±0.122 | 0.422±0.170 | 1.891±0.673 | 1.064±0.211 |
| | Stem | 0.679±0.111 | 0.574±0.124 | 0.415±0.173 | 1.799±0.613 | 1.045±0.205 |
| | HiST (Ours) | **0.716±0.117** | **0.641±0.137** | **0.504±0.195** | **1.464±0.564** | **0.951±0.197** |
| Skin | TRIPLEX | 0.826±0.093 | 0.795±0.112 | 0.745±0.136 | 0.957±0.488 | 0.675±0.219 |
| | StNet | 0.808±0.100 | 0.782±0.113 | 0.739±0.135 | 0.970±0.402 | 0.677±0.192 |
| | BLEEP | 0.782±0.108 | 0.755±0.120 | 0.708±0.139 | 1.117±0.559 | 0.696±0.224 |
| | Stem | 0.781±0.097 | 0.749±0.116 | 0.698±0.140 | 1.196±0.626 | 0.707±0.241 |
| | HiST (Ours) | **0.838±0.089** | **0.812±0.104** | **0.766±0.127** | **0.882±0.387** | **0.631±0.181** |
| Kidney | TRIPLEX | 0.541±0.094 | 0.453±0.087 | 0.331±0.068 | 1.138±0.255 | 0.843±0.098 |
| | StNet | 0.537±0.111 | 0.454±0.100 | 0.332±0.079 | 1.084±0.153 | 0.822±0.052 |
| | BLEEP | 0.508±0.123 | 0.426±0.114 | 0.306±0.087 | 1.181±0.219 | 0.853±0.080 |
| | Stem | 0.501±0.112 | 0.404±0.097 | 0.273±0.067 | 1.288±0.199 | 0.888±0.071 |
| | HiST (Ours) | **0.618±0.099** | **0.530±0.090** | **0.399±0.071** | **1.079±0.198** | **0.821±0.082** |

## C.5 Evaluation with Patient-Level Data Splitting

We implemented a strict patient-level data splitting protocol to address potential data leakage and ensure a more rigorous evaluation of generalization performance. This ensures that all Whole Slide Images (WSIs) from a single patient are confined to a single data partition (train, validation, or test), adopting a stricter and more clinically meaningful evaluation standard than prior works (Zhu et al., 2025; Xie et al., 2023).

As shown in Table 8, this more challenging setup led to a performance drop across all methods, which confirms the presence of patient-specific features and highlights the importance of this strict separation. Crucially, even under these stringent conditions, HiST maintains its position as the top-performing model. It consistently achieves state-of-the-art results across most Pearson Correlation (PCC) metrics and datasets, demonstrating that its architectural advantages provide greater generalization capability to new, unseen patients, making it more robust for real-world applications.

Table 8: Performance comparison on three spatial transcriptomics datasets using strict patient-level splits. **Bold** indicates the best performance, and underline indicate the second-best performance.

| Dataset | Model | PCC@10 ↑ | PCC@50 ↑ | PCC@200 ↑ | MSE ↓ | MAE ↓ |
|---|---|---|---|---|---|---|
| Colorectum | TRIPLEX | 0.685±0.115 | 0.613±0.128 | 0.418±0.116 | 2.423±0.744 | 1.187±0.187 |
| | StNet | 0.654±0.111 | 0.585±0.114 | 0.386±0.091 | 2.284±0.319 | 1.155±0.098 |
| | BLEEP | 0.612±0.119 | 0.538±0.117 | 0.340±0.081 | 2.551±0.513 | 1.213±0.137 |
| | Stem | 0.653±0.136 | 0.560±0.126 | 0.357±0.086 | 2.351±0.504 | 1.166±0.144 |
| | HiST | **0.712±0.096** | **0.642±0.093** | **0.448±0.078** | **2.152±0.515** | **1.131±0.152** |
| Skin | TRIPLEX | 0.816±0.089 | 0.783±0.108 | 0.726±0.131 | 1.361±0.701 | 0.812±0.196 |
| | StNet | 0.786±0.102 | 0.759±0.112 | 0.704±0.128 | 1.407±0.652 | 0.832±0.194 |
| | BLEEP | 0.789±0.102 | 0.761±0.116 | 0.705±0.132 | 1.479±0.627 | 0.826±0.204 |
| | Stem | 0.765±0.103 | 0.734±0.117 | 0.673±0.132 | 1.743±1.040 | 0.898±0.264 |
| | HiST | **0.828±0.079** | **0.799±0.093** | **0.746±0.111** | **1.310±0.727** | **0.792±0.210** |
| Kidney | TRIPLEX | 0.511±0.107 | 0.418±0.109 | 0.288±0.102 | **1.207±0.249** | **0.868±0.100** |
| | StNet | 0.503±0.079 | 0.425±0.079 | 0.303±0.077 | 1.332±0.311 | 0.917±0.166 |
| | BLEEP | 0.495±0.083 | 0.407±0.085 | 0.279±0.086 | 1.368±0.365 | 0.915±0.137 |
| | Stem | 0.468±0.073 | 0.365±0.063 | 0.227±0.059 | 1.449±0.323 | 0.942±0.115 |
| | HiST | **0.587±0.065** | **0.495±0.069** | **0.357±0.074** | 1.283±0.362 | 0.889±0.140 |

## C.6 SCALABILITY WITH NUMBER OF TARGET GENES

We investigated the model's scalability by analyzing how its performance, memory usage, and training time scale with an increasing number of target genes. As shown in Table 9, increasing the number of target genes from 200 to 3000 has a minimal impact on memory and training time. This is because only the first and last linear layers of the gene encoder and decoder are affected by the number of genes. While the computational cost of these layers scales linearly, this overhead is insignificant compared to the model's overall complexity, demonstrating HiST's excellent scalability for larger-scale genomic analyses.

Table 9: Analysis of performance, memory, and training time scale with the number of target genes in the HiST Model on the HCC dataset.

| Metric | Target Genes | | |
|---|---|---|---|
| | 200 | 1000 | 3000 |
| PCC@10 ↑ | 0.721±0.105 | 0.723±0.114 | 0.713±0.107 |
| PCC@50 ↑ | 0.642±0.128 | 0.655±0.133 | 0.649±0.123 |
| PCC@200 ↑ | 0.477±0.184 | 0.557±0.165 | 0.564±0.153 |
| PCC@500 ↑ | - | 0.472±0.187 | 0.492±0.167 |
| PCC@1000 ↑ | - | 0.387±0.184 | 0.429±0.169 |
| PCC@2000 ↑ | - | - | 0.353±0.156 |
| PCC@3000 ↑ | - | - | 0.143±0.294 |
| MSE ↓ | 1.498±0.456 | 0.933±0.215 | 0.632±0.095 |
| MAE ↓ | 0.958±0.158 | 0.766±0.092 | 0.621±0.039 |
| Memory Usage (GB) | 18.39 | 18.43 | 18.76 |
| Training Time (s/epoch) | 74 | 73 | 73 |
| Inference Time (s/epoch) | 3 | 3 | 2 |

## C.7 SENSITIVITY TO NEIGHBORHOOD SIZE

The construction of a niche is dependent on the neighborhood size, a key hyperparameter in our model. To evaluate the model's sensitivity to this hyperparameter, we varied the neighborhood sizes (6, 18, and 36) for niche construction on the HCC dataset. As shown in Table 10, the model's performance remains highly stable across the different settings. This demonstrates that HiST is robust and not sensitive to the choice of neighborhood size.

Table 10: Performance comparison on different neighborhood sizes on the HCC dataset.

| Neighborhood sizes | PCC@10 ↑ | PCC@50 ↑ | PCC@200 ↑ | MSE ↓ | MAE ↓ |
|---|---|---|---|---|---|
| 6 | 0.721±0.105 | 0.642±0.128 | 0.477±0.184 | 1.498±0.456 | 0.958±0.158 |
| 18 | 0.714±0.112 | 0.633±0.132 | 0.474±0.173 | 1.569±0.582 | 0.978±0.189 |
| 36 | 0.716±0.109 | 0.636±0.126 | 0.474±0.172 | 1.616±0.613 | 0.993±0.201 |

## C.8 QUANTITATIVE GENE-WISE ANALYSIS

Our quantitative, per-gene analysis reveals that the enhanced performance of HiST on spatially-constrained genes (e.g., immune, structural, and tissue-specific markers) validates our hypothesis that morphological context provides meaningful constraints for these gene categories. Table 11 presents a detailed comparison of the gene-wise Pearson Correlation Coefficient (PCC) improvement of HiST over TRIPLEX in the tumor region of a representative sample, highlighting significant performance gains on genes with known biological importance.

Table 11: Gene-wise PCC improvement of HiST vs. TRIPLEX in the tumor region of sample ZEN42. Genes with significant biological importance are highlighted in **bold**.

| Gene | PCC (HiST) | PCC (TRIPLEX) | PCC Improvement |
|---|---|---|---|
| **HLA-DQB1** | 0.592 | 0.237 | 0.355 |
| **S100A14** | 0.575 | 0.224 | 0.351 |
| **GREM1** | 0.625 | 0.296 | 0.329 |
| FABP1 | 0.161 | -0.166 | 0.327 |
| **PIGR** | 0.810 | 0.511 | 0.299 |
| MUC5B | 0.580 | 0.291 | 0.289 |
| **REG3A** | 0.536 | 0.279 | 0.257 |
| **REG1A** | 0.476 | 0.226 | 0.250 |
| **LCN2** | 0.497 | 0.258 | 0.240 |
| FXYD3 | 0.461 | 0.230 | 0.232 |
| **HLA-DPB1** | 0.539 | 0.315 | 0.223 |
| **TFF3** | 0.769 | 0.552 | 0.217 |
| **CD74** | 0.589 | 0.379 | 0.210 |
| SPINK1 | 0.531 | 0.346 | 0.186 |

## C.9 BIOLOGICAL VALIDATION ON KEY MARKER GENES

To demonstrate the translational value and biological insights of our method, we conducted a region-specific gene prediction analysis targeting the tumor-stromal interface in colorectal cancer, a critical area for prognosis and therapeutic response (Valdeolivas et al., 2024; Feng et al., 2024). We selected three clinically validated marker genes representing distinct functional compartments: **KRT18** (epithelial tumor core), **ACTA2** (stromal activation), and **IGKC** (B-cell/plasma cell immune response).

As shown in Table 12, we measured the region-specific prediction accuracy improvement over baseline methods. The results indicate that HiST more accurately captures the spatial expression patterns of these critical genes, which is essential for downstream applications such as therapeutic target identification, immune infiltration assessment, and tumor margin evaluation.

## D ADDITIONAL VISUALIZATION

In this section, we present visualizations of additional marker genes across different tissues to demonstrate the robustness of our findings. We selected three samples (MICS27, NCBI476, and NCBI704) from datasets of lung, skin, and kidney tissues, respectively, and plotted the predicted gene expressions on their H&E images for the following marker genes: ALDH1A1, ELF3, GJB2, KRT6B, UMOD, and PODXL. These genes are highly correlated with specific tissue cell types. Specifically,

Table 12: Region-specific gene prediction accuracy improvement over baseline models. P-values were determined by a paired Student's t-test across tissue regions.

| Gene | Model | MAE Improv. (%) | MSE Improv. (%) | Correlation Improv. (%) |
|------|-------|-----------------|-----------------|-------------------------|
| **KRT18** | BLEEP | 8.80 (0.1084) | 23.13 (0.0698) | 18.92 |
| | iSTAR | 52.42 ($<$0.001) | 131.37 ($<$0.001) | 12.91 |
| | Stem | 17.94 ($<$0.001) | 42.71 ($<$0.001) | 7.56 |
| | StNet | 9.86 (0.0875) | 31.95 (0.0171) | 14.04 |
| | TRIPLEX | 84.57 ($<$0.001) | 238.58 ($<$0.001) | 10.40 |
| **ACTA2** | BLEEP | 36.66 (0.0064) | 70.55 (0.0582) | 5.34 |
| | iSTAR | 43.89 (0.0016) | 108.59 (0.0105) | 24.04 |
| | Stem | 10.67 (0.4406) | 16.43 (0.6867) | 2.45 |
| | StNet | 28.76 (0.0414) | 62.14 (0.1025) | 1.08 |
| | TRIPLEX | 36.89 (0.0413) | 72.50 (0.1153) | 7.72 |
| **IGKC** | BLEEP | 53.18 (0.0037) | 231.16 (0.0169) | 41.84 |
| | iSTAR | 4.26 (0.8164) | 52.25 (0.4332) | 1.96 |
| | Stem | 53.06 (0.0030) | 250.32 (0.0072) | 31.50 |
| | StNet | 36.73 (0.0299) | 206.47 (0.0417) | 32.97 |
| | TRIPLEX | 143.21 ($<$0.001) | 691.59 ($<$0.001) | 24.43 |

ALDH1A1 and ELF3 are marker genes for lung tissue, GJB2 and KRT6B are marker genes for skin tissue, and UMOD and PODXL are marker genes for kidney tissue. Visualizations of predicted gene expressions by different models, along with their PCC comparisons, are presented in Figures 7 to 10. The figures clearly demonstrate that HiST's gene expression predictions exhibit strong consistency with the ground truth, achieving higher PCC than existing models.

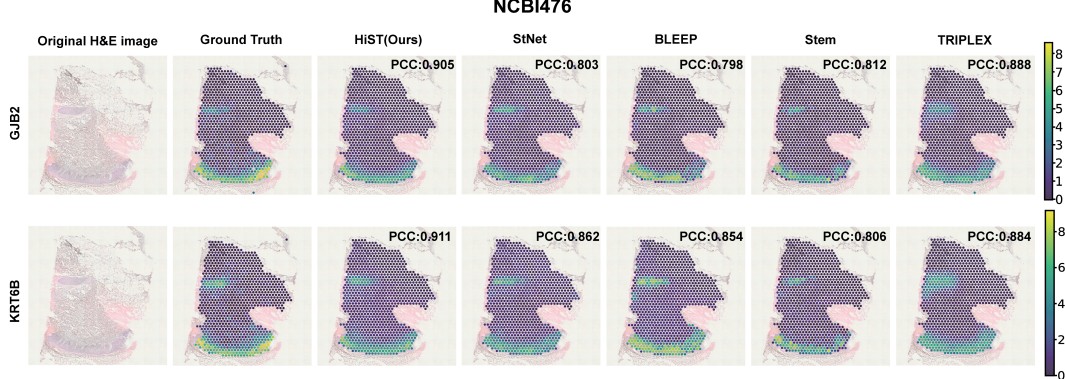

Figure 7: Visualization of GJB2 and KRT6B gene predictions in the NCBI476 sample from the Skin dataset. We present the PCC values comparing the ground truth with the gene expression predictions generated by each model.

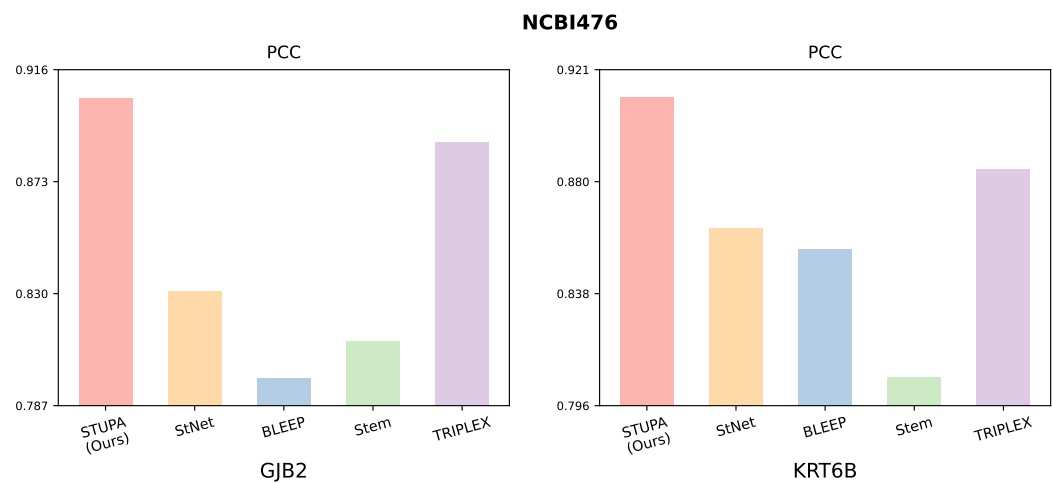

Figure 8: PCC comparison of GJB2 and KRT6B gene predictions (Higher PCC reflects greater accuracy)

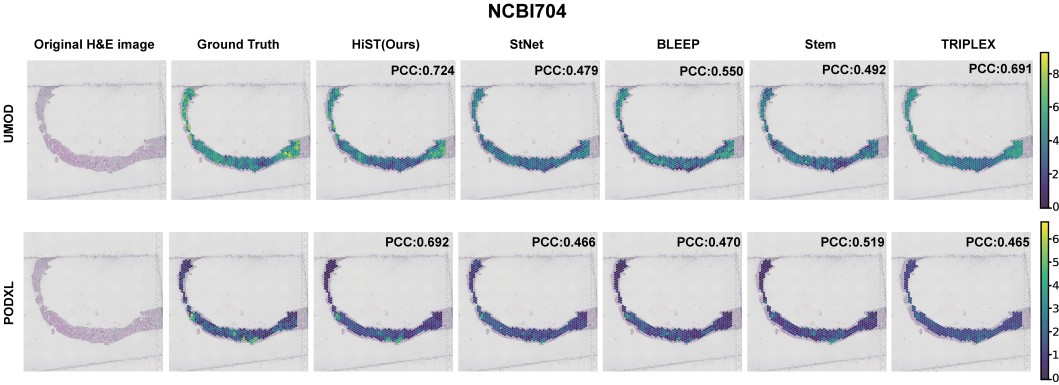

Figure 9: Visualization of UMOD and PODXL gene predictions in the NCBI704 sample from the Kidney dataset. We present the PCC values comparing the ground truth with the gene expression predictions generated by each model.

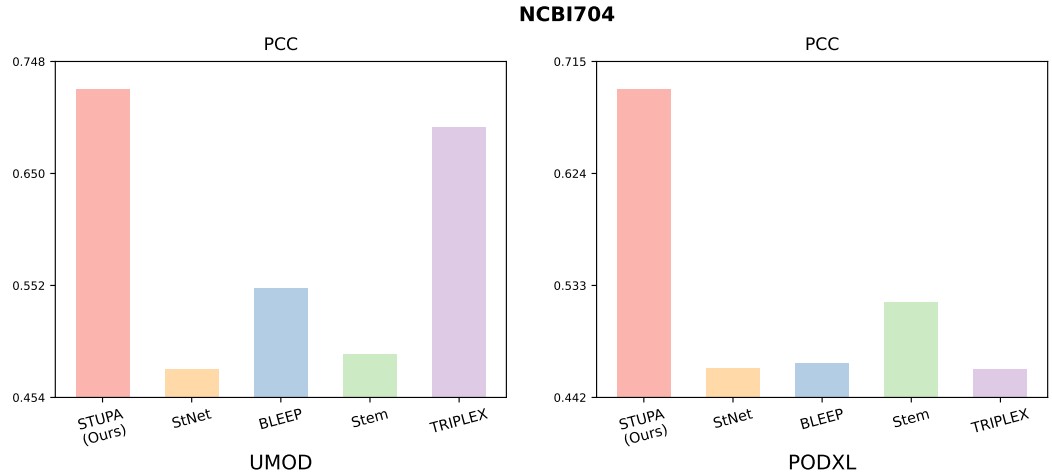

Figure 10: PCC comparison of UMOD and PODXL gene predictions (Higher PCC reflects greater accuracy)

# E  ADDITIONAL ABLATION STUDY

We conduct further ablation studies on the HMHVG gene set for the Skin and Kidney datasets, with results for alignment strategy, LoRA settings, and decoder input detailed in Tables 13, 14, and 15, respectively. Through these ablation studies, we rigorously evaluate the contribution of HiST's individual modules to system performance. These experiments demonstrate that HiST consistently achieves optimal performance in the majority of experimental conditions, highlighting the effectiveness and robustness of these modules.

Table 13: Additional ablation study of the alignment strategy

| Dataset | Alignment | PCC@10 ↑ | PCC@50 ↑ | PCC@200 ↑ | MSE ↓ | MAE ↓ |
|---|---|---|---|---|---|---|
| Skin | w/o G-I HEA | 0.833±0.084 | 0.805±0.100 | 0.753±0.126 | 0.981±0.431 | 0.672±0.184 |
| | w/o HEA | 0.832±0.090 | 0.803±0.108 | 0.746±0.136 | 0.974±0.418 | 0.681±0.181 |
| | w/o HEA + HCA | 0.809±0.099 | 0.783±0.113 | 0.731±0.137 | 1.006±0.437 | 0.695±0.187 |
| | MERU | 0.812±0.098 | 0.784±0.115 | 0.730±0.142 | 0.986±0.420 | 0.672±0.181 |
| | CLIP | 0.810±0.098 | 0.784±0.113 | 0.733±0.137 | 0.980±0.393 | 0.681±0.176 |
| | Ours | 0.839±0.086 | 0.812±0.102 | 0.758±0.129 | 0.932±0.418 | 0.657±0.182 |
| Kidney | w/o G-I HEA | 0.610±0.101 | 0.514±0.096 | 0.378±0.074 | 1.147±0.188 | 0.839±0.065 |
| | w/o HEA | 0.607±0.092 | 0.513±0.085 | 0.377±0.066 | 1.131±0.195 | 0.837±0.074 |
| | w/o HEA + HCA | 0.576±0.099 | 0.484±0.089 | 0.344±0.064 | 1.134±0.168 | 0.837±0.058 |
| | MERU | 0.586±0.099 | 0.494±0.090 | 0.355±0.062 | 1.148±0.205 | 0.842±0.071 |
| | CLIP | 0.558±0.098 | 0.462±0.087 | 0.321±0.058 | 1.220±0.293 | 0.867±0.097 |
| | Ours | 0.617±0.094 | 0.526±0.088 | 0.390±0.07 | 1.077±0.155 | 0.817±0.058 |

Table 14: Ablation study of the LoRA settings

| Dataset | LoRA Layers | PCC@10 ↑ | PCC@50 ↑ | PCC@200 ↑ | MSE ↓ | MAE ↓ |
|---|---|---|---|---|---|---|
| Colorectum | 0 | 0.701±0.126 | 0.622±0.151 | 0.458±0.200 | 1.662±0.638 | 1.007±0.212 |
| | 5 | 0.704±0.131 | 0.619±0.158 | 0.456±0.201 | 1.565±0.612 | 0.976±0.207 |
| | 7 | 0.713±0.117 | 0.627±0.146 | 0.468±0.187 | 1.562±0.606 | 0.978±0.209 |
| | Ours (11) | 0.721±0.105 | 0.642±0.128 | 0.477±0.184 | 1.498±0.456 | 0.958±0.158 |
| Skin | 0 | 0.831±0.088 | 0.804±0.104 | 0.751±0.130 | 0.977±0.437 | 0.677±0.185 |
| | 5 | 0.833±0.088 | 0.806±0.105 | 0.754±0.130 | 0.971±0.438 | 0.670±0.189 |
| | 7 | 0.838±0.083 | 0.810±0.100 | 0.756±0.126 | 0.944±0.413 | 0.654±0.184 |
| | Ours (11) | 0.839±0.086 | 0.812±0.102 | 0.758±0.129 | 0.932±0.418 | 0.657±0.182 |
| Kidney | 0 | 0.587±0.099 | 0.498±0.088 | 0.366±0.071 | 1.110±0.162 | 0.828±0.059 |
| | 5 | 0.603±0.096 | 0.510±0.088 | 0.376±0.071 | 1.100±0.207 | 0.826±0.083 |
| | 7 | 0.606±0.105 | 0.511±0.100 | 0.378±0.084 | 1.106±0.193 | 0.827±0.077 |
| | Ours (11) | 0.617±0.094 | 0.526±0.088 | 0.390±0.070 | 1.077±0.155 | 0.817±0.058 |

Table 15: Additional ablation study of the decoder input

| Dataset | Decoder Input | PCC@10 ↑ | PCC@50 ↑ | PCC@200 ↑ | MSE ↓ | MAE ↓ |
|---|---|---|---|---|---|---|
| Colorectum | only spot | 0.709±0.098 | 0.622±0.115 | 0.450±0.169 | 1.564±0.334 | 0.973±0.119 |
| | only niche | 0.661±0.179 | 0.570±0.203 | 0.405±0.238 | 1.801±0.878 | 1.031±0.268 |
| | Ours (spot+niche) | 0.721±0.105 | 0.642±0.128 | 0.477±0.184 | 1.498±0.456 | 0.958±0.158 |
| Skin | only spot | 0.814±0.096 | 0.787±0.111 | 0.735±0.134 | 1.003±0.417 | 0.673±0.187 |
| | only niche | 0.831±0.092 | 0.803±0.107 | 0.748±0.132 | 0.979±0.441 | 0.670±0.188 |
| | Ours (spot+niche) | 0.839±0.086 | 0.812±0.102 | 0.758±0.129 | 0.932±0.418 | 0.657±0.182 |
| Kidney | only spot | 0.584±0.093 | 0.492±0.088 | 0.353±0.067 | 1.171±0.184 | 0.849±0.065 |
| | only niche | 0.588±0.097 | 0.491±0.090 | 0.356±0.073 | 1.110±0.139 | 0.828±0.051 |
| | Ours (spot+niche) | 0.617±0.094 | 0.526±0.088 | 0.390±0.070 | 1.077±0.155 | 0.817±0.058 |

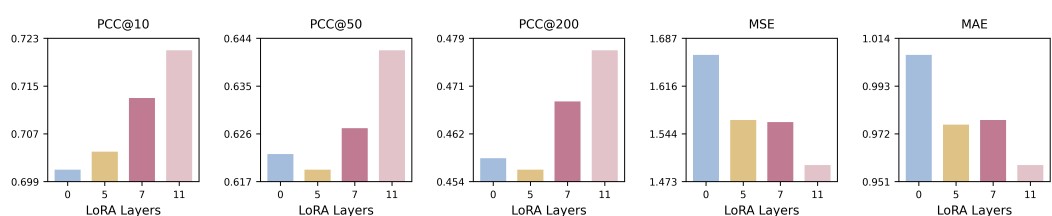

Figure 11: Ablation study on the choice of the last layers of LoRA for Colorectum

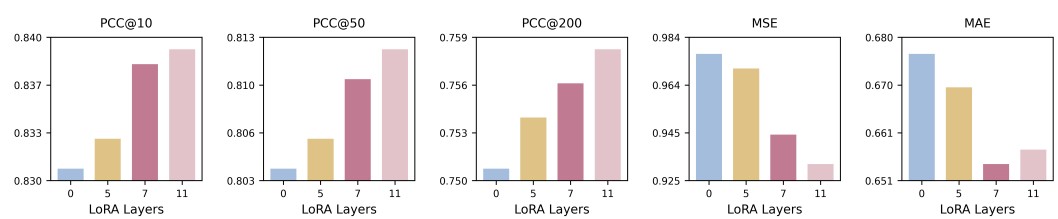

Figure 12: Ablation study on the choice of the last layers of LoRA for Skin

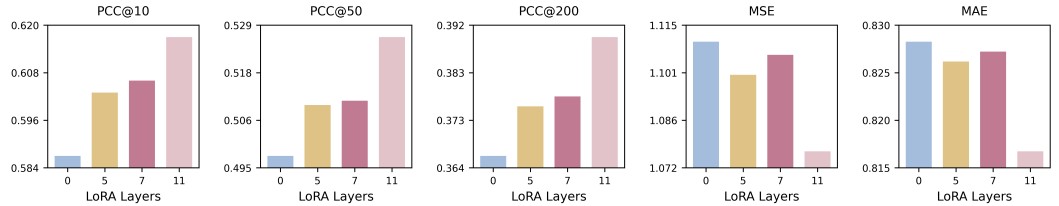

Figure 13: Ablation study on the choice of the last layers of LoRA for Kidney

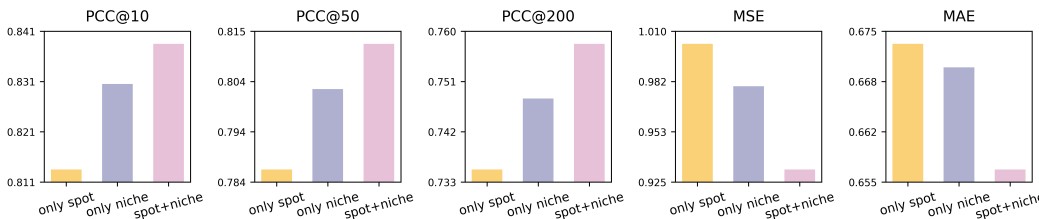

Figure 14: Ablation study of the input data of decoder for Skin

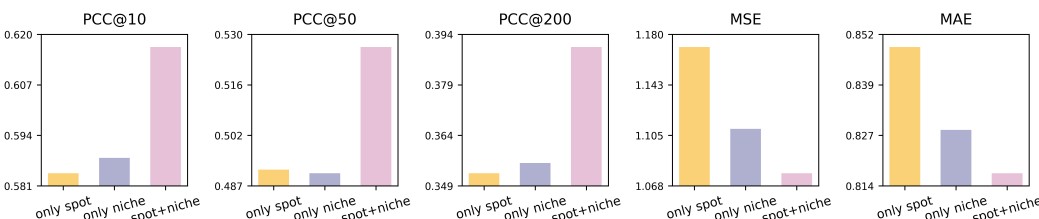

Figure 15: Ablation study of the input data of decoder for Kidney

## F  LIMITATION

While HiST advances spatial transcriptomics prediction through the hierarchical hyperbolic structure, several limitations merit discussion. First, the gene decoder relies on a simple multilayer perceptron (MLP) to predict expression from aligned image representations, potentially limiting its capacity to model complex gene-gene interactions. Second, the integration of multi-level features (spot-level and niche-level images) is achieved through concatenation, which may overlook fine-grained feature fusion. Third, the model is trained and fine-tuned on a limited dataset, and its generalization capability could be further enhanced by training on larger, more diverse cohorts. In the future, we plan to address these limitations by training on a large-scale dataset and transforming HiST into a foundation model for broader applicability.

## G  THE USE OF LARGE LANGUAGE MODELS (LLMS)

In the writing of this paper, we used large language models only for aiding and polishing the text. We did not use them for research ideation or to make a substantive contribution to the content of the paper.

## H  BROADER IMPACT

### H.1  IMPACT ON REAL-WORLD APPLICATIONS

For decades, WSIs have served as a foundation in biomedical research and clinical diagnostics. However, the expensive and labor-intensive nature of Spatial Transcriptomics (ST) limits its broad use, driving the demand for deep learning methods that directly predict spatially resolved gene expression from these images. To tackle the challenges of incorporating broader pathological and genetic contexts while effectively capturing target-modality information, we present HiST, an innovative framework for predicting spatial transcriptomics. HiST leverages multi-level hyperbolic image-gene representations, which model hierarchical structures and improve the integration of cross-modal features. Evaluated across three diverse tissue datasets, HiST consistently surpasses current state-of-the-art models, showcasing its performance in predicting spatial gene expression. Therefore, the proposed method, which reduces research costs and enhances efficiency, shows promise in advancing the development and application of spatial transcriptomics. The primary focus of our work is to advance scientific methodology, and we foresee no direct negative societal consequences.

### H.2  IMPACT ON FUTURE RESEARCH

In this paper, we observe that current spatial transcriptomics prediction methods, which operate within traditional Euclidean space, neglect the inherent multi-level structure of gene expression and overlook biological heterogeneity. Given that hyperbolic space is more suitable for modeling hierarchical structures and can capture richer information, it enhances the cross-modal representation between pathological images and gene expression. Therefore, we propose HiST, which, in contrast to traditional Euclidean approaches, incorporates Multi-Level Representation Extractors and Hierarchical Hyperbolic Alignment to fully exploit the intrinsic hierarchical structure of ST data. We hope this framework will inspire future studies in multimodal learning that leverage geometric structures, such as hyperbolic spaces, in the biological field, like single-cell RNA sequencing, proteomics, and other data modalities. Blending the inherent geometry of biological systems with multimodal learning can guide the development of future sophisticated representation models.

