# OpenReview forum: "HiST: Spatial Transcriptomics Prediction via Multi-Level Hyperbolic Representation Learning"
_ICLR.cc/2026/Conference — ICLR 2026 Conference Withdrawn Submission_

### Official Review · Reviewer_B8HJ · 2025-10-31

**Soundness:** 2
**Presentation:** 2
**Contribution:** 2
**Rating:** 6
**Confidence:** 3

**Summary:**

The paper focuses on predicting gene expression from pathology images by modeling spatial transcriptomics. To leverage the hierarchical structure of spatial transcriptomics data, it learns multi-level image-gene features by embedding the inherent hierarchy of the data within hyperbolic space. A Multi-Level Representation Extractor is proposed to capture both spot-level and niche-level features, while a Hierarchical Hyperbolic Alignment module is introduced to perform spatial alignment among these features. Experiments on three tissue types (colorectum, skin, and kidney) demonstrate the effectiveness of the proposed method.

**Strengths:**

- The paper presents a strong motivation for learning hierarchical representation features from pathology images.
- The paper demonstrates strong performance on established datasets.

**Weaknesses:**

- Unclear effectiveness of hierarchical representation features

The paper addresses the general challenge of learning hierarchical representation features from both pathology images and their corresponding gene expression profiles. However, the only downstream task evaluated is gene expression prediction, with minimal mention of MSI status classification. This makes it difficult to fully assess whether the learned features of pathology images are truly hierarchical. Additionally, HCA and HEA function more like pre-training objectives. It remains unclear how the method would perform if the training were separated into two distinct stages: pre-training and downstream fine-tuning. Furthermore, there are alternative objectives for encoding global and local (hierarchical) features such as DINOV2 and it would be informative to compare the performance of these methods in this context.

- Experimental setting

Why were experiments not conducted on Visium HD, which provides multi-level gene expression data that better aligns with the paper’s goal of learning hierarchical representation features?

The experiments are limited to colorectum, skin, and kidney from the HEST-1K dataset. It is unclear why other subsets were not considered, and whether the chosen tissues are representative enough to support general conclusions.

**Questions:**

- Relating to L366

This raises a potential bias in the method. Pathology images are continuous at the pixel level, and what defines a “spot” is determined by how gene expression is measured, not by intrinsic image boundaries. Using K-nearest neighbors to define a niche might artificially impose a discrete structure on inherently continuous data. It is unclear why the authors chose K-nearest neighbors instead of simply expanding the context around the central spot, which could achieve a similar effect without introducing potential bias.

---

### Official Review · Reviewer_ynvx · 2025-10-31

**Soundness:** 2
**Presentation:** 2
**Contribution:** 2
**Rating:** 2
**Confidence:** 3

**Summary:**

This paper proposes HiST, a framework for predicting spatially-resolved gene expression from histology images using hyperbolic representation learning. The authors used Hierarchical structure modeling to model the spot-level to niche-level contexts in hyperbolic space. HiST improves expression prediction across three public datasets using HEST1k and validates clinical utility.

**Strengths:**

1. The authors have clearly presented the core idea in Figure 2.
2. The task of predicting gene expression from H&E has high impact.

**Weaknesses:**

1. The core theoretical foundation that ST data forms hierarchies based on "information specificity" (line 96) is unconvincing:
   1. From a spot, you cannot infer the environmental/biological gradients or neighborhood context (eg, proximity to vasculature, immune infiltration from adjacent tissue, tumor boundaries). From a niche, you cannot determine spot-specific cellular heterogeneity (eg, single-cell composition, microenvironmental niches within the spot). This is a complementary relationship where each level provides unique information. I'm not seeing a parent-child hierarchy where one entails the other
   2. Similarly, morphology lacks information about genes without morphological correlates. Gene expression also lacks information about: spatial organization and tissue-level patterns invisible at the molecular level.
2. Following last comment, the ablation is also not supporting the use of hyperbolic representation alignment. The error bars of `w/o G-I HEA ` ,`w/o HEA `, `w/o HEA + HCA ` heavily overlap and the means show a bare difference for pearson across # of genes (table 13)
3. It is unclear how the MSI status prediction is done, is the input to RF pseudo bulk? It's unclear how to interpret the results without a reference to how GT gene expression can do as well.
4. HiST seems to be very compute inefficient with 2x more training time compared with the second slowest method (Table 5)

**Questions:**

See Weaknesses

---

### Official Review · Reviewer_7bmd · 2025-11-01

**Soundness:** 2
**Presentation:** 3
**Contribution:** 2
**Rating:** 4
**Confidence:** 4

**Summary:**

The authors introduce a method for predicting gene expression from H&E images using hyperbolic geometry to model hierarchical ST data relationships. The framework combines multi-level extractors (spot/niche features) with hierarchical hyperbolic alignment, validated on three datasets with modest improvements over baselines.

**Strengths:**

- The motivation for treating gene expression as child concepts of images in an entailment hierarchy is conceptually interesting, given genes contain richer molecular information.
- Authors provided ablation experiments examining alignment strategies, LoRA configurations, and decoder inputs. Patient-level splitting addresses data leakage concerns appropriately.
- MSI prediction on TCGA-COADREAD demonstrates some translational potential beyond just reconstruction metrics.

**Weaknesses:**

- The justification for hyperbolic geometry is weak—no proof shows why this hierarchy (spot to niche, image to gene) requires hyperbolic over Euclidean space. The information specificity argument is vague, and ST data exhibits many-to-many relationships that contradict the tree-like structure hyperbolic space was designed for.
- The empirical gains are marginal given the complexity cost. Improvements over Euclidean CLIP baseline are small, while 7.4× training time vs BLEEP. Several comparisons are also non significant, undermining claims of consistent improvement.
- The authors conflate multi-level features with hyperbolic geometry benefits. Authors demonstrate pot+niche combination helps regardless of geometry, and TRIPLEX achieves second-best performance using multi-scale features in Euclidean space. The paper fails to properly disentangle whether gains come from hierarchical features or hyperbolic space.
- Critical experimental validations are missing or insufficient. Cross-laboratory test uses only one WSI, no comparison to recent foundation models (Virchow, Prov-GigaPath), TCGA experiment excludes key baselines, and zero visualization of learned hyperbolic embeddings to validate the claimed hierarchical organization. Gene selection is restrictive, and Table 9 shows poor scaling.
- Technical details lack justification and analysis. Trainable curvature c, entailment parameter K=0.1, and loss weights α=0.2, β=0.4 have no sensitivity analysis or selection rationale.
- Lorentzian distance in contrastive loss is unbounded but impact on temperature parameter and gradient flow is never discussed. Gene-level results show mixed outcomes suggesting potential overfitting.

**Questions:**

- How will the method perform on genes with low spatial autocorrelation where niche context provides minimal information?
- Can the authors show t-SNE/UMAP of hyperbolic embeddings demonstrating the claimed spot to niche & image to gene hierarchies?
- How does performance compare to Euclidean space with explicit tree-structured losses instead of hyperbolic geometry?

---

### Official Review · Reviewer_1MY5 · 2025-11-09

**Soundness:** 2
**Presentation:** 3
**Contribution:** 3
**Rating:** 6
**Confidence:** 5

**Summary:**

This work introduces HiST, a method for predicting spatial gene expression from histology images by learning multi‑level image–gene representations in hyperbolic space. HiST works by first extracting both spot‑level and neighborhood (or "niche") features from each modality so each spot’s signal is contextualized by its local tissue environment, then applies hyperbolic representation learning for spatial alignment of image and gene embeddings. Experiments compare HiST against prior image‑to‑gene prediction baselines (TRIPLEX, STNet, BLEEP, Stem) on three ST datasets from HEST-1k. Ablations include multimodal alignment strategy (without their proposed hyperbolic alignment strategy, comparison against CLIP and MERU which represent the conventional contrastive and hyperbolic alignment strategies from image-text), inputs to gene decoder, and LoRA.

**Strengths:**

- Introducing hyperbolic representation learning for pathology-ST integration is a unique idea that hasn't been approached before. Proposed loss objectives ("Hierarchical Contrastive Alignment" and "Hierarchical Entailment Alignment") I found to be quite unique and interesting for modeling spot-niche hierarchies.
- Experimental design is overall quite sound, and presentation of technical details are well-written and clear. Appendix is also organized with detailed discussion of experiments, ablation experiments, and evaluation methodology. This work is towards the top of my stack in terms of works that make good technical contributions.

**Weaknesses:**

- Why were only 3 organs in HEST-1k evaluated?
- I would like to see an additional comparison to OmiCLIP [1], a recent work on pathology-ST pretraining.

References
1. Chen, W., Zhang, P., Tran, T.N., Xiao, Y., Li, S., Shah, V.V., Cheng, H., Brannan, K.W., Youker, K., Lai, L. and Fang, L., 2025. A visual–omics foundation model to bridge histopathology with spatial transcriptomics. Nature Methods, pp.1-15.

**Questions:**

N/A

---

### Note · Authors · 2025-11-12

I have read and agree with the venue's withdrawal policy on behalf of myself and my co-authors.